# Sudden Drops in the Loss: Syntax Acquisition, Phase Transitions, and Simplicity Bias in MLMs

**Angelica Chen**[1]    **Ravid Shwartz-Ziv**[1]    **Kyunghyun Cho**[1,2,3]
**Matthew L. Leavitt**[4]    **Naomi Saphra**[5]

{angelica.chen, ravid.shwartz.ziv, kyunghyun.cho}@nyu.edu
matthew@datologyai.com   nsaphra@fas.harvard.edu
[1]NYU   [2]Genentech   [3]CIFAR LMB   [4] DatologyAI   [5] Kempner Institute, Harvard

## Abstract

Most interpretability research in NLP focuses on understanding the behavior and features of a fully trained model. However, certain insights into model behavior may only be accessible by observing the trajectory of the training process. We present a case study of syntax acquisition in masked language models (MLMs) that demonstrates how analyzing the evolution of interpretable artifacts throughout training deepens our understanding of emergent behavior. In particular, we study Syntactic Attention Structure (SAS), a naturally emerging property of MLMs wherein specific Transformer heads tend to focus on specific syntactic relations. We identify a brief window in pretraining when models abruptly acquire SAS, concurrent with a steep drop in loss. This breakthrough precipitates the subsequent acquisition of linguistic capabilities. We then examine the causal role of SAS by manipulating SAS during training, and demonstrate that SAS is necessary for the development of grammatical capabilities. We further find that SAS competes with other beneficial traits during training, and that briefly suppressing SAS improves model quality. These findings offer an interpretation of a real-world example of both simplicity bias and breakthrough training dynamics.

## 1 Introduction

While language model training usually leads to smooth improvements in loss over time (Kaplan et al., 2020), not all knowledge emerges uniformly. Instead, language models acquire different capabilities at different points in training. Some capabilities remain fixed (Press et al., 2023), while others decline (McKenzie et al., 2022), as a function of dataset size or model capacity. Certain capabilities even exhibit abrupt improvements—this paper focuses on such discontinuous dynamics, which are often called **breakthroughs** (Srivastava et al., 2022), **emergence** (Wei et al., 2022), **breaks** (Caballero et al., 2023), or **phase transitions** (Olsson et al., 2022). The interpretability literature rarely illuminates how these capabilities emerge, in part because most analyses only examine the final trained model. Instead, we consider *developmental* analysis as a complementary explanatory lens.

To better understand the role of interpretable artifacts in model development, we analyze and manipulate these artifacts during training. We focus on a case study of **Syntactic Attention Structure** (SAS), a model behavior thought to relate to grammatical structure. By measuring and controlling the emergence of SAS, we deepen our understanding of the relationship between the internal structural traits and extrinsic capabilities of masked language models (MLMs).

SAS occurs when a model learns specialized attention heads that focus on a word's syntactic neighbors. This behavior emerges naturally during conventional MLM pre-training (Clark et al., 2019; Voita et al., 2019; Manning et al., 2020). We observe an abrupt spike in SAS at a consistent point in training, and explore its impact on MLM capabilities by manipulating SAS during training. Our observations paint a picture of how interpretability artifacts may represent simplicity biases that compete with other learning strategies during MLM training. In summary, our main contributions are:

- Monitoring latent syntactic structure (defined in Section 2.1) throughout training, we identify (Section 4.1) a precipitous loss drop composed of multiple phase transitions (defined in Section 2.3)

relating to various linguistic abilities. At the onset of this stage (which we call the **structure onset**), SAS spikes. After the spike, the model starts handling complex linguistic phenomena correctly, as signaled by a break in BLiMP score (which we call the **capabilities onset**). Although the functional complexity of the model declines for the rest of training, it increases between these breaks.

- We introduce a regularizer to examine the causal role of SAS (defined in Section 2.2) and use it to show that SAS is necessary for handling complex linguistic phenomena (Section 4.2) and that SAS competes with an alternative strategy that exhibits its own break in the loss curve, which we call the **alternative strategy onset**.
- Section 4.3 shows that briefly suppressing SAS improves model quality and accelerates convergence. Suppressing past the alternative strategy onset damages performance and blocks SAS long-term, suggesting this phase transition terminates a critical learning period.

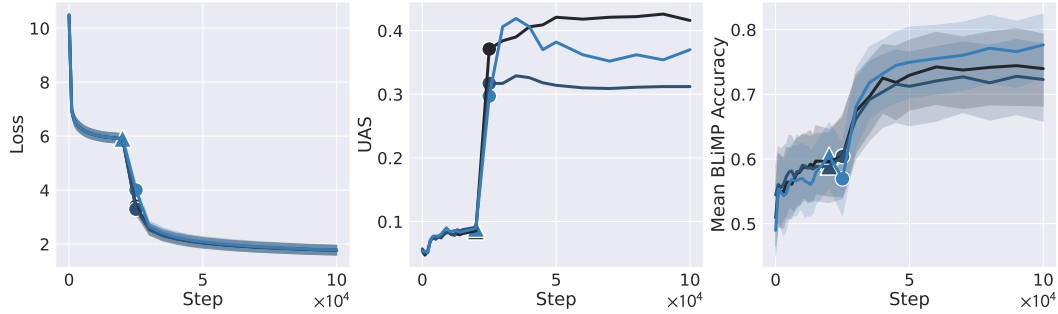

Figure 1: **BERT first learns to focus on syntactic neighbors with specialized attention heads, and then exhibits grammatical capabilities in its MLM objective**. The former (internal) and the latter (external) model behaviors both emerge abruptly, at moments we respectively call the **structure onset** (▲) and **capabilities onset** (●) (quantified as described in Section 2.3). We separately visualize three runs with different seeds, noting that these seeds differ in the stability of Unlabeled Attachment Score (UAS; see Section 2.1) after the structure onset, but uniformly show that SAS emerges almost entirely in a brief window of time. We show (a) MLM loss, with 95% confidence intervals across samples bynonparametric bootstrapping; (b) internal grammar structure, measured by UAS on the parse induced by the attention distributions; and (c) external grammar capabilities, measured by average BLiMP accuracy with 95% confidence intervals across tasks by nonparametric bootstrapping.

## 2 METHODS

### 2.1 SYNTACTIC ATTENTION STRUCTURE

One proposal for interpreting attention is to treat some attention weights as syntactic connections (Manning et al., 2020; Voita et al., 2019; Clark et al., 2019). Our method is based on Clark et al. (2019), who find that some specialized attention heads focus on the target word's dependency relations.

Dependency parses describe latent syntactic structure. Each word in a sentence has a word that it modifies, which is its parent in the syntax tree. Each dependency is labeled—e.g., an adjective modifies a noun through an `amod` relation in the Universal Dependencies annotation system (Nivre et al., 2017). In the example that follows, when an MLM predicts the word *nests*, it is likely to rely heavily on its syntactic relations *builds* and *ugly*. One head may attend to adjectival modifiers like *ugly* while another attends to direct objects like *builds*.

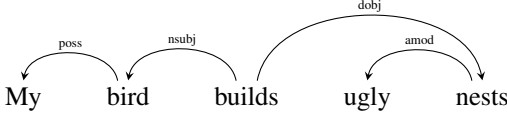

We call this tendency to form heads that specialize in specific syntactic relations Syntactic Attention Structure (SAS). To measure SAS, we follow Clark et al. (2019) in using a simple probe based off

the surface-level attention patterns, detailed in Appendix A. The probe provides an implicit parse, with an accuracy measured by **unlabeled attachment score** (UAS).

## 2.2 CONTROLLING SAS

In addition to training models with BERT$_{\text{Base}}$ parameters, we also train models where SAS is promoted or suppressed. The model with SAS promoted throughout training is called BERT$_{\text{SAS+}}$, while the model with SAS suppressed throughout training is called BERT$_{\text{SAS-}}$.

In order to adjust SAS for these models, we train a BERT$_{\text{Base}}$ model through methods that are largely conventional (Section 3.1), with one difference. We add a **syntactic regularizer** that manipulates the structure of the attention distributions using a syntacticity score $\gamma(x_i, x_j)$, equal to the maximum attention weight between syntactically connected words $i$ and $j$. We use this regularizer to penalize or reward higher attention weights on a token's syntactic neighbors by adding it to the MLM loss $L_{\text{MLM}}$. We scale the regularizer by a constant coefficient $\lambda$ which may be negative to promote SAS or positive to suppress SAS. If we denote $D(x)$ as the set of all dependents of $x$, then the new loss is:

$$L(x) = \underbrace{L_{\text{MLM}}(x)}_{\text{Original loss}} + \underbrace{\lambda \sum_{i=1}^{|x|} \sum_{x_j \in D(x_i)} \gamma(x_i, x_j)}_{\text{Syntactic regularization}} \tag{1}$$

## 2.3 IDENTIFYING BREAKTHROUGHS

This paper studies *breakthroughs*: sudden changes in model behavior during a brief window of training. We use the term *breakthroughs* interchangeably with *phase transitions*, *breaks*, and *emergence*, as has been done in past literature (Olsson et al., 2022; Srivastava et al., 2022; Wei et al., 2022; Caballero et al., 2023). What do we consider to be a breakthrough, given a metric $f$ at some distance (e.g., in timesteps) from initialization $d$? We are looking for break point $d^*$ with the sharpest angle in the trajectory of $f$, as determined by the slope between $d^*$ and $d^* \pm \Delta$ for some distance $\Delta$. If we have no measurements at the required distance, we infer a value for $f$ based on the available checkpoints—e.g., if $d$ is measured in discrete timesteps, we calculate the angle of loss at 50K steps for $\Delta = 5K$ by imputing the loss from checkpoints at 45K and 55K steps to calculate slope.

$$\text{break}(f, \Delta) = \arg \max_t \left[ f(t + \Delta) - f(t) \right] - \left[ f(t) - f(t - \Delta) \right] \tag{2}$$

In other words, break$(f, \Delta)$ is the point $t$ that maximizes the difference between the slope from $f(t)$ to $f(t + \Delta)$ and the slope from $f(t - \Delta)$ to $f(t)$, approximating the point of maximum acceleration.

## 3 MODELS AND DATA

### 3.1 ARCHITECTURE AND TRAINING

We pre-train BERT$_{\text{Base}}$ models using largely the same training set-up and dataset as Sellam et al. (2022). We use the uncased architecture with 12 layers of 768 dimensions each and train with the AdamW optimizer (Loshchilov & Hutter, 2019) for 1M steps with learning rate of 1e-4, 10,000 warm-up steps and training batch size of 256 on a single $4 \times 100$ NVIDIA A100 node. Our results only consider checkpoints that are recorded while pretraining remains numerically stable for all seeds, so we only analyze up to 300K steps.

Our training set-up departs from the original BERT set-up (Devlin et al., 2019) in that we use a fixed sequence length of 512 throughout training, which was shared by Sellam et al. (2022). We also use the same WordPiece-based tokenizer as Devlin et al. (2019) and mask tokens with 15% probability. Unless otherwise stated, all experiments are implemented with the HuggingFace transformers (v4.12.5) (Wolf et al., 2020), Huggingface datasets (v2.7.1) (Lhoest et al., 2021), and Pytorch (v1.11) (Paszke et al., 2019) libraries.

Our pre-training datasets consist of BookCorpus (Zhu et al., 2015) and English Wikipedia (Foundation, 2022). Since we do not have access to the original BERT pre-training dataset, we use a more recent Wikipedia dump from May 2020. For pre-training runs where syntactic regularization is applied, we dependency parse Wikipedia with `spacy` (Honnibal & Montani, 2017) for our silver standard labels.

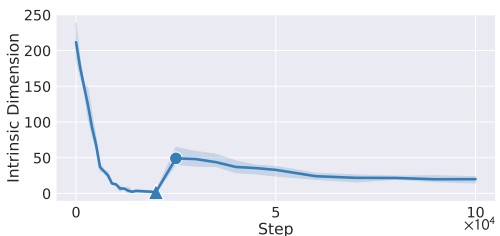

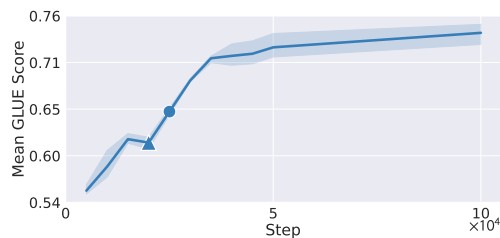

(a) Intrinsic dimension (TwoNN) complexity. Alternative complexity metrics in Appendix L.2.

(b) Average finetuned GLUE score. Individual tasks in Appendix J.

Figure 2: Metrics during BERT$_{\text{Base}}$ training averaged, with 95% confidence intervals, across three seeds. Structure (▲) and capabilities (●) onsets are marked.

## 3.2 FINETUNING AND PROBING

**Fine-tuning on GLUE**   Our fine-tuning set-up for each GLUE task matches that of the original paper (Wang et al., 2018), with initial learning rate 1e-4, batch size of 32, and 3 total epochs.

**Evaluating on BLiMP**   BLiMP (Warstadt et al., 2020a) is a benchmark of minimal pairs for evaluating knowledge of various English grammatical phenomena. We evaluate performance using the MLM scoring function from Salazar et al. (2020) to compute the pseudo-log-likelihood of the sentences in each minimal pair, and counting the MLM as correct when it assigns a higher value to the acceptable sentence in the pair. Further implementation details are in Appendix D.

**Evaluating SAS dependency parsing**   We measure SAS by evaluating the model's implicit best-head attention parse (Eq. (3), Clark et al., 2019) on a random sample of 1000 documents from the Wall Street Journal portion of the Penn Treebank (Marcus et al., 1999), with the syntax annotation labels converted into the Stanford Dependencies format by the Stanford Dependencies converter (Schuster & Manning, 2016). We evaluate parse quality using the **Unlabeled Attachment Score** (UAS) computed from the attention map, as described in Eq. (3).

## 4 RESULTS

Often, interpretable artifacts are assumed to be essential to model performance. However, evidence for the importance of SAS exists only at the instance level on a single trained model. We know that specialized heads can predict dependencies (Clark et al., 2019) and that pruning them damages performance more than pruning other heads (Voita et al., 2019). However, these results are only weak evidence that SAS is essential for modeling grammar. Passive observation of a trained model may discover artifacts that occur as a side effect of training without any effect on model capabilities. Causal methods that intervene on particular components at test time (Vig et al., 2020; Meng et al., 2023), meanwhile, may interact with the rest of the model in complex ways, spuriously implying a component to be essential for performance when it could be removed if it were not so entangled with other features. They also only address whether a component is necessary at test time, and not whether that component is necessary during *learning*. Both test-time approaches—passive observations and causal interventions—are limited.

We begin by confirming the assumption that SAS must be essential to performance. To motivate the case for skepticism of the role of SAS, we note a lack of correlation between SAS metrics and model capabilities across random pretraining seeds (Appendix E). After first strengthening the evidence for SAS as a meaningful phenomenon by taking model development into account, we then draw connections to the literature on phase transitions, simplicity bias, and model complexity.

## 4.1 THE SYNTAX ACQUISITION PHASE

Most work on scaling laws (Kaplan et al., 2020) presents test loss as a quantity that homogeneously responds to the scale of training, declining by a power law relative to the size of the corpus. In

the MLM setting, we instead identify a precipitous drop in the loss curve of $BERT_{Base}$ (Fig. 1(a)), consistently spanning 20K-30K timesteps of training across various random seeds. We now show how this rapid learning stage can be interpreted as the composition of two distinct phase transitions.

*The MLM loss drop occurs alongside the acquisition of grammatical capabilities in two consecutive stages*, each distinguished by breaks as defined by Eq. (2). The first stage aligns with the formation of SAS—we call this break in implicit parse UAS the **structure onset**. As seen in Fig. 1(b), the UAS spikes at a consistent time during each run, in tandem with abrupt improvements in MLM loss (Fig. 1(a)) and finetuning metrics (Fig. 2(b)). Immediately following the spike, UAS plateaus, but the loss continues to drop precipitously before leveling off. The second part of this loss drop is associated with a break in the observed grammatical capabilities of the model, as measured by accuracy on BLiMP (Fig. 1(c)). We call the BLiMP break the **capabilities onset**. We show similar trajectories on the MultiBERTs (Sellam et al., 2022) reproductions (Appendix F).

By observing these phase transitions, we can see that the *internal* representation of grammar, in the form of syntactic attention, precipitates the *external* observation of grammatical behavior, in the form of correct language modeling judgements on linguistically challenging examples. This is not only a single breakthrough during training, but a sequence of breakthroughs that appear to be dependent on each other. We might compare this to the "checkmate in one" BIG-Bench task, a known breakthrough behavior in autoregressive language models (Srivastava et al., 2022). Only at a large scale can models accurately identify checkmate moves, but further exploration revealed that the model was progressing in a linear fashion at offering consistently valid chess moves before that point. The authors posited that the checkmate capability was dependent on the ability to make valid chess moves, and likewise it seems we have found that grammatical capabilities are dependent on a latent representation of syntactic structure in the form of SAS.

We find that the existence of these phase transitions holds even when using continuous metrics (Appendix I), in contrast to Schaeffer et al. (2023), who found that many abrupt improvements in capabilities are due to the choice of thresholded metrics like accuracy. We also find that the phase transitions hold even when setting the x-axis to some continuous alternative to discrete training timesteps, such as weight norm (Appendix H). Thus both x-axis and y-axis may use non-thresholded scales, and the phase transitions remain present.

**Complexity and Compression**  The capabilities of language models have often been explained as a form of compression (Chiang, 2023; Cho, 2023; Sutskever, 2023), implying a continual decrease in functional complexity throughout training. A more nuanced view of training is suggested by work on critical learning periods (Achille et al., 2018) and the information bottleneck (IB) theory (Shwartz-Ziv & Tishby, 2017b), which tie phase transitions to shifts in complexity (Arnold et al., 2023). When we plot various complexity metrics (see Fig. 2(a) for TwoNN intrinsic dimension (Facco et al., 2017), and Appendix L.2 for additional metrics), we see a decline in complexity for most of training. However, an abrupt **memorization phase** of increasing complexity occurs between the structure and capabilities onsets, during which the model rapidly acquires information. (Although the mid-training timing of this memorization phase is novel, these dual phases are supported by the literature on IB and critical learning periods, as further discussed in Appendix G.) Taken together, we see that the structure and capabilities onsets are not just phase transitions of internal structure and external capabilities, but also phase transitions of functional complexity.

The steep decrease in complexity that immediately precedes the structure onset (Fig. 2(a)) supports the intuitive understanding that SAS must be simplistic in order to be human interpretable. Generally, models tend to favor simpler functions like SAS earlier in training (Hermann & Lampinen, 2020; Shah et al., 2020; Nakkiran et al., 2019; Valle-Pérez et al., 2019; Arpit et al., 2017), a tendency often referred to as **simplicity bias**. In Section 4.3 we use the simplicity bias literature to motivate our hypotheses about and interventions for competing strategies.

## 4.2  CONTROLLING SAS

Having established the natural emergence of SAS, we use our syntacticity regularizer (Section 2.2) to evaluate whether SAS is truly necessary for handling complex grammatical phenomena. We confirm that this regularizer can suppress or accelerate the SAS phase (Fig. 3(b)). As seen in Fig. 3(a), *enhancing* SAS behavior throughout training ($BERT_{SAS+}$) leads to early improvements in MLM

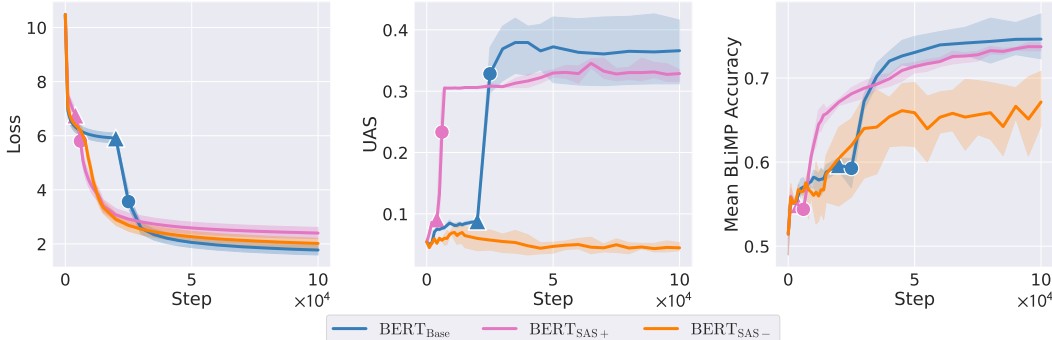

Figure 3: Metrics over the course of training for baseline and SAS-regularized models (under both suppression and promotion of SAS). Structure (▲) and capabilities (●) onsets are marked, except on BERT$_{SAS-}$, which does not clearly exhibit either onset. Each line is averaged over three random seeds. On y-axis: (a) MLM loss (b) Implicit parse accuracy (c) average BLiMP accuracy over all phenomena categories. Shaded regions represent the 95% confidence interval.

performance, but hurts later model quality.[1] Conversely, *suppressing* SAS (BERT$_{SAS-}$) damages both early performance and long term performance. Suppressing SAS during training prevents the emergence of linguistically complex capabilities (Fig. 3(c)). In other words, preventing the internal grammar structure onset will also prevent the external grammar capabilities onset that follows it.

However, there exists an early apparent phase transition in the MLM loss (at around 6K steps), which suggests that an alternative strategy emerges that leads to improvements prior to the structure onset. We therefore refer to this early inflection as the **alternative strategy onset**. Our results suggest that SAS is crucial for effectively representing grammar, but the existence of the alternative strategy onset implies that SAS also competes with other useful traits in the network. We explore the alternative strategy onset represented by the phase transition under SAS suppression in Appendix M.

Importantly, the break in the loss curve occurs earlier in training when suppressing SAS. The implication is profound: that the alternative strategy is competing with SAS, and suppressing SAS permits the model to learn the alternative strategy more effectively and earlier. Inspired by this insight, we next ask whether there can be larger advantages to avoiding the natural SAS-based strategy early in training, thus claiming the benefits of the alternative strategy.

## 4.3 EARLY-STAGE SAS REGULARIZATION

Because BERT$_{SAS-}$ briefly outperforms both BERT$_{Base}$ and BERT$_{SAS+}$, we have argued that suppressing SAS implicitly promotes a competing strategy. This notion of competition between features or strategies is well-documented in the literature on simplicity bias (Shah et al., 2020; Arpit et al., 2017; Hermann & Lampinen, 2020; Pezeshki et al., 2021). Achille et al. (2018) find that some patterns must be acquired early in training in order to be learned at all, so depending too much on an overly simplistic strategy early in training can have significant long-term consequences on performance. To test the hypothesis that learning SAS early allows SAS to out-compete other beneficial strategies, this section presents experiments that only suppress the early acquisition of SAS. For multistage regularized models, we first suppress SAS with $\lambda = 0.001$ and then set $\lambda = 0$ after a pre-specified timestep in training. These models are named after the timestep that SAS is suppressed until, e.g., BERT$_{SAS-}^{(3k)}$ is the model where $\lambda$ is set to 0 at timestep 3000.

We find that suppressing SAS early on improves the effectiveness of training later. Specifically, BERT$_{SAS-}^{(3k)}$ outperforms BERT$_{Base}$ even well after both models pass their structure and capabilities onsets (Fig. 4(a); Table 1), although these advantages cease to be significant after longer training runs

---

[1]Note that in BERT$_{SAS+}$, we see the capabilities onset is after the structure onset, but before the SAS plateau, suggesting that SAS only needs to hit some threshold to precipitate the capabilities onset, and does not need to stabilize.

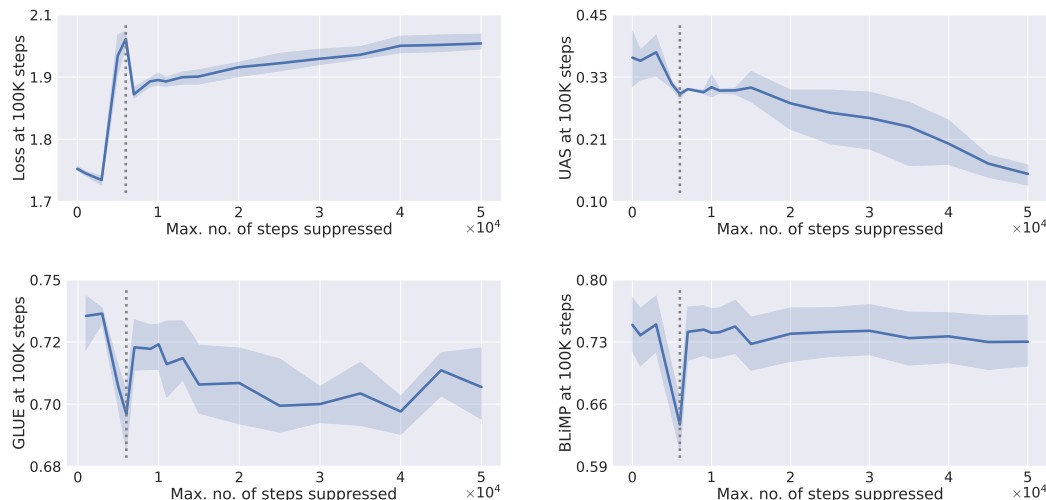

Figure 4: Metrics for the checkpoint at 100k steps, for various models with SAS suppressed early in training. The vertical line marks the BERT$_{SAS-}$ alternative strategy onset; note that *model quality is worst when the regularizer is changed during this phase transition*. The x-axis reflects the timestep when the regularizer $\lambda$ is changed from $0.001$ to $0$. To control for the length of training time without suppressing SAS, Appendix O presents the same findings measured at a checkpoint exactly 50K timesteps after releasing the regularizer. On y-axis: (a) MLM loss; (b) Implicit parse accuracy (UAS); (c) GLUE average (Task breakdown in Appendix J); (d) BLiMP average (Task break down in Appendix K). Shaded regions represent 95% confidence intervals across three seeds.

| | MLM Loss $\downarrow$ | GLUE average $\uparrow$ | BLiMP average $\uparrow$ |
|---|---|---|---|
| BERT$_{Base}$ | $1.77 \pm 0.01$ | $\mathbf{0.74 \pm 0.01}$ | $\mathbf{0.74 \pm 0.02}$ |
| BERT$_{SAS+}$ | $2.39 \pm 0.03$ | $0.59 \pm 0.01$ | $\mathbf{0.74 \pm 0.01}$ |
| BERT$_{SAS-}$ | $2.02 \pm 0.01$ | $0.69 \pm 0.02$ | $0.67 \pm 0.03$ |
| BERT$_{SAS-}^{(3k)}$ | $\mathbf{1.75 \pm 0.01}$ | $\mathbf{0.74 \pm 0.00}$ | $\mathbf{0.75 \pm 0.01}$ |

Table 1: Evaluation metrics, with standard error, after training for 100K steps ($\sim 13$M tokens), averaged across three random seeds for each regularizer setting. We selected BERT$_{SAS-}^{(3k)}$ as the best multistage hyperparameter setting based on MLM test loss at 100K steps. Bolded values significantly outperform non-bolded values in the same column under a 1-sided Welch's $t$-test.

(Appendix P). Some multistage models even have more consistent SAS than BERT$_{Base}$ (Fig. 4(b)). We posit that certain associative patterns are learned more quickly while suppressing SAS, and these patterns not only support overall performance but even provide improved features to acquire SAS.

### 4.3.1 WHEN CAN WE RECOVER THE SAS PHASE TRANSITION?

Inspecting the learning curves of the temporarily suppressed models, we find that briefly suppressing SAS can promote performance (Appendix N) and accelerate the structure onset (Fig. 5(a)) while augmenting it (Fig. 5(b)). However, after more prolonged suppression of SAS, it becomes impossible to hit the dramatic spike in implicit parse UAS that we see in BERT$_{Base}$ (Section 4.3). If the SAS phase transition is prevented, MLM performance falls significantly compared to BERT$_{Base}$ and we see no SAS spike (Appendix N). It appears that we must choose between phase transitions; *the model cannot undergo first the alternative strategy onset and then the structure onset*. In fact, we measure the worst model quality when we switch settings *during* the alternative strategy transition (Fig. 4).

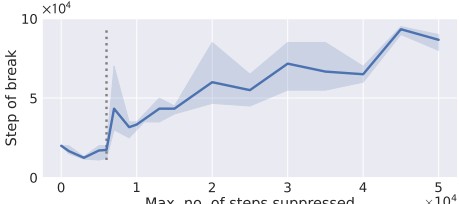

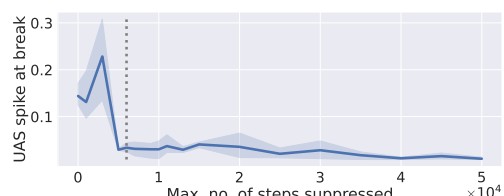

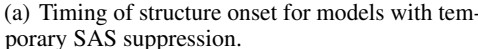

(a) Timing of structure onset for models with temporary SAS suppression.

(b) Magnitude of the UAS spike at structure onset time $t$, determined by $\text{UAS}_{t+5000} - \text{UAS}_t$.

Figure 5: If SAS is suppressed only briefly, it accelerates and augments the SAS onset. However, further suppression delays and attenuates the spike in UAS, until it eventually ceases to show a clear inflection. A vertical dotted line marks the $\text{BERT}_{\text{SAS-}}$ alternative strategy onset and the shaded region indicates the 95% confidence interval across three seeds.

## 5 DISCUSSION AND CONCLUSIONS

Our work is a response to the limitations of probes that analyze only a single model checkpoint without regard to its training history (Saphra, 2023). We posit that *developmental* explanations, which incorporate a model's training history, provide critical perspective and explanatory power. We have used this developmental approach to demonstrate the necessity of SAS for grammatical reasoning in MLMs, and have furthermore used SAS as a case study to shed light on alleviating simplicity bias, the dynamics of model complexity, and the risks of changing the optimizer during phase transitions. Our work also guides further understanding of many deep learning phenomena, and may inspire a more rigorous approach to science of deep learning. For further discussion Appendix C presents an extended review of related work on causal interpretability, simplicity bias, and phase transitions.

### 5.1 EARLY DYNAMICS AND SIMPLICITY BIAS

Sutton (2019) introduced the machine learning world to the *Bitter Lesson*: models that use informed priors based on domain understanding will always lose to generic models trained on large quantities of data. Our work suggests that we might go further: even generically learned structure can form a disadvantageously strong prior, if that structure reflects human expert models of syntax. In other words, human interpretations of natural phenomena are so simplistic that their presence early in training can serve as a negative signal. If this observation holds in natural language—a modality that has evolved specifically to be human interpretable—how much worse might simplicity bias impact performance on other domains like scientific and physical modeling?

**Dependency and competition**  We have found evidence of multiple possible relationships between emergent behaviors. Previous work suggests that model properties can *depend* on one another, e.g., checkmate-in-one capabilities depend on first learning valid chess moves (Srivastava et al., 2022); or *compete* with one another, e.g., as sparse and dense representation strategies compete on arithmetic tasks (Merrill et al., 2023). Similarly, we first find evidence of a dependency relationship, based on our evidence that SAS is a prerequisite for many linguistic capabilities as indicated by BLiMP. Then, we identify a competitive relationship, based on our observations that suppressing SAS leads to an alternative strategy that prioritizes context differently. These distinct relationships shed light on how model behaviors interact during training and may suggest training improvements that delay or promote particular behaviors. Existing work in simplicity bias (Shah et al., 2020; Pezeshki et al., 2021) suggests that a preference for simple heuristics might prevent the model from acquiring a more reliable strategy. Our results appear to be evidence of this pitfall in practice.

**Pretraining**  The benefits of early training without permitting SAS bear an intriguing parallel to pretraining. Just as pretraining removes the particulars of the downstream task by training on generic language structure, early SAS suppression removes the particulars of linguistic structure itself. In doing so, we encourage the MLM to treat the entire sequence without regard for proximity to the target word, as a bag-of-words model might. Therefore, the beginning of training is even more unstructured and generic than it would be under the baseline MLM objective.

**Curriculum learning**  We also offer some insights into why curriculum learning is rarely effective at large scales. Simple data is likely to encourage simplistic strategies, so any curriculum that homogenizes the early distribution could promote a simplistic strategy, helping early performance but harming later performance. Predictably, curricula no longer help at large scales (Wu et al., 2021).

## 5.2   PHASE TRANSITIONS

**Instability at critical points**  Abrupt changes are rarely documented directly at the level of validation loss, but we show that they may be observed—and interpreted—in realistic settings. Smooth improvements in loss may even elide abrupt breakthroughs in specific capabilities, as discussed in Appendix M.1. Our multistage results point to a surprising effect: that the worst time to change the regularization is during a phase transition. When we release the SAS suppression well *before* the point at which the alternative transition starts during $BERT_{SAS}$- training (i.e., the alternative strategy onset), we find it is possible to recover the SAS transition, preventing damage to GLUE, BLiMP, and MLM loss metrics. Likewise, although releasing the regularization *after* the alternative transition prevents the recovery of SAS, it nonetheless incurs limited damage to model quality metrics. However, releasing the regularizer *during* the phase transition leads to a substantially worse model under every metric. These findings suggest that the moment of breakthrough constitutes a critical point where an optimizer misstep can damage the performance of the model, possibly even at convergence. This phenomenon may be consequential for future optimization research.

## 5.3   INTERPRETABILITY EPISTEMOLOGY

While SAS was already known to emerge naturally in MLMs, there were reasons to be skeptical of its necessity. One objection is that raw attention distribution information is not a guaranteed proxy for information flow (Abnar & Zuidema, 2020; Ethayarajh & Jurafsky, 2021). Another thread questions the interpretability of attention by obfuscating the attention weights without damaging model performance (Jain & Wallace, 2019; Serrano & Smith, 2019). If the fundamentally informative nature of attention is subject to extensive debate (Bibal et al., 2022), we must also be skeptical of overstating its connection to syntax. Attention syntacticity is a microcosm of wider failures in the science of deep learning, which has been criticized for a tendency to use anecdotal observations and post-hoc explanations, rather than rigorous correlational or causal tests (Forde & Paganini, 2019).

Prior evidence for the importance of SAS came in two forms, both of which operate post-hoc at the instance level on specific samples: instance-level observation in fully trained networks (Clark et al., 2019) and instance-level causal experiments in fully trained networks (Voita et al., 2019). Observational studies might discover structures that emerge as a side effect of training, rather than those crucial to the operation of the model. Traits that emerge as a side effect of a process but appear crucial to performance are called *spandrels* in evolutionary biology; possible examples include human chins (Yong, 2016) and enjoyment of music (Pinker, 1997). While instance-level causal experiments like Voita et al. (2019) may be epistemically stronger than the observational studies, the network's failure to recover from a causal intervention does not indicate that it relies on the structure provided. Instead, the network may be more brittle to large distribution shifts on the relevant features, without truly relying on those features (Tucker et al., 2021). One possible scenario is that a behavior may develop early in training and become *vestigial* (like a human's tailbone (Mukhopadhyay et al., 2012)) but sufficiently integrated into subnetworks that generate and cancel information that the network cannot easily recover from its removal. To support the skeptical case, we find that SAS metrics were not correlated with MLM capabilities across random seed (Fig. 6).

We provide several epistemically strong results in favor of the importance of SAS. First, we study models in development (Section 4.1), finding that the SAS phase transition directly precipitates the emergence of linguistic capabilities. This result supports that blackbox grammatical capabilities depend on measurable internal structures. Second, we have causal interventions on development (Section 4.2), which again reveal the importance of this head specialization behavior by promoting and suppressing it. Instance-level interpretability methods, at best, offer evidence that a trait emerges and the model cannot recover from its removal; we can now say that certain capabilities depend on this trait—although the model eventually discovers alternative ways to represent some of them.

ACKNOWLEDGMENTS

We thank Samuel R. Bowman and Jason Phang for helpful discussions during the development of this project.

This work was supported by National Science Foundation Award 1922658, Hyundai Motor Company (under the project Uncertainty in Neural Sequence Modeling), and the Samsung Advanced Institute of Technology (under the project Next Generation Deep Learning: From Pattern Recognition to AI). Matthew Leavitt was employed at Mosaic ML and Naomi Saphra was employed at NYU for the majority of the work on this project.

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

## A  SYNTACTIC ATTENTION STRUCTURE PROBE

We use the following simple probe, based on (Clark et al., 2019), to measure syntactic attention structure. First, we define a head-specific probe $f_{h,l}$ that predicts the parent word for a target word $x_i$ from the attention map $\alpha$ in head $h$ and layer $l$. Denoting $\alpha_{ij}^{(h,l)}$ as the attention weight between words $i$ and $j$ for attention head $h$ in layer $l$, we define this probe as:

$$f_{h,l}(x_i) := \arg\max_{x_j} \left[ \max \left( \alpha_{ij}^{(h,l)}, \alpha_{ji}^{(h,l)} \right) \right], \tag{3}$$

In other words, given target word $i$ and attention head $h$ in layer $l$, $f_{h,l}$ predicts the other word that receives the maximum attention across both directions (e.g., from parent to child and child to parent). Since BERT$_{\text{Base}}$ uses byte-pair tokenization (Sennrich et al., 2016), we convert the token-level attention maps to word-level attention maps. Attention to a word is summed over its constituent tokens, and attention from a word is averaged over its tokens, as in Clark et al. (2019).

However, this probe is head-specific. To then acquire a parent predictor for a given dependency relation, we select the best-performing head (as determined by accuracy of the predicted parent words when compared against silver labels) for each relation. Denoting the set of all dependency relations as $\mathcal{R}$ and each relation $R \in \mathcal{R}$ as the set of all ordered word pairs $(x, y)$ that have that relation (with $y$ being the parent of $x$), the best-performing probe for each relation $R$ is:

$$\hat{h}_R = \arg\max_{f_{h,l}} \frac{1}{|R|} \sum_{x \in \mathcal{D}} \sum_{(x_i, x_j) \in R} \mathbb{1}_R \left( (x_i, f_{h,l}(x_i)) \right) \tag{4}$$

where $\mathcal{D}$ is the dataset, $x_i$ and $x_j$ represent the $i$-th and $j$-th words in example $x$, and $\mathbb{1}_R$ is the indicator function for set $R$—i.e., $\mathbb{1}_R \left( (x_i, f_{h,l}(x_i)) \right) = 1$ if $(x_i, f_{h,l}(x_i)) \in R$ (which occurs only when $f_{h,l}(x_i) = x_j$ since each word can only have one parent) and 0 otherwise. Lastly, we use these relation-specific probes to compute the overall accuracy across all dependency relations. The resulting accuracy is known as the **Unlabeled Attachment Score** (UAS):

$$\text{UAS} := \frac{1}{\sum_{R \in \mathcal{R}} |R|} \sum_{R \in \mathcal{R}} \sum_{x \in \mathcal{D}} \sum_{(x_i, x_j) \in R} \mathbb{1}_R \left( (x, \hat{h}_R(x)) \right) \tag{5}$$

We compute UAS on a random sample of 1000 documents from the Wall Street Journal portion of the Penn Treebank (Marcus et al., 1999) and use Stanford Dependencies (Schuster & Manning, 2016) parses as our silver labels of which word pairs $(x_i, x_j)$ correspond to each dependency relation $R \in \mathcal{R}$.

## B  SYNTACTIC REGULARIZER

We add a **syntactic regularizer** that manipulates the structure of the attention distributions. The regularizer adds a syntacticity score $\gamma(x_i, x_j)$ that is equal to the maximum attention weight (summed across the forward and backward directions) between words $i$ and $j$, for all pairs $(i, j)$ where there exists some dependency relation between $i$ and $j$. Because heads tend to specialize in particular syntactic relations, we compute this maximum over all heads and layers. More precisely,

$$\gamma(x_i, x_j) = \max_{h,l} \alpha_{ij}^{(h,l)} + \max_{h,l} \alpha_{ji}^{(h,l)} \tag{6}$$

We use this regularizer to either penalize or reward higher attention weights on a token's syntactic neighbors by adding it to the MLM loss $L_{\text{MLM}}$, scaled by a constant coefficient $\lambda$. We set $\lambda < 0$ and $\lambda > 0$ to promote and suppress syntacticity, respectively. If we denote $\text{parent}(x)$ as the dependency parent of $x$ and $D(x) := \{y \mid x = \text{parent}(y)\}$ as the set of all dependents of $x$, then the entire loss objective is:

$$L(x) = \underbrace{L_{\text{MLM}}(x)}_{\text{Original loss}} + \underbrace{\lambda \sum_{i=1}^{|x|} \sum_{x_j \in D(x_i)} \gamma(x_i, x_j)}_{\text{Syntactic regularization}} \tag{7}$$

# C RELATED WORK

Our work links two parallel bodies of literature, one from the interpretability community, focusing on causal methods of interpretation; and the other from the training dynamics community, focusing on phase changes and on the influence of early training. We combine insights from both communities, studying the role of interpretable artifacts on model generalization by measuring both variables while intervening on training.

## C.1 SIMPLICITY BIAS AND PHASE CHANGES

Models tend to learn simpler functions earlier in training (Hermann & Lampinen, 2020; Shah et al., 2020; Nakkiran et al., 2019; Valle-Pérez et al., 2019; Arpit et al., 2017). In LMs, Choshen et al. (2022) identify a trend in BLiMP (Warstadt et al., 2020a) grammatical tests during training: earlier on, LMs behave like n-gram language models, but later in training they diverge. Likewise, LMs learn early representations that are similar to representations learned for simplified versions of the language modeling task, like part-of-speech prediction (Saphra & Lopez, 2019). Despite gradual increases in complexity, SGD exhibits a bias towards simpler functions and features that are already learned (Pezeshki et al., 2021), so simplistic functions learned early in training can still shape the decisions of a fully trained model. Importantly, a large degree of simplicity bias can be disadvantageous to robustness, calibration, and accuracy (Shah et al., 2020), which inspires our approach of limiting access to interpretable—and thus simplistic, as interpretable behaviors must be simple enough to understand (Lipton, 2018)—solutions early in training.

In studying transitions between simplistic internal heuristics and more complex model behavior, we incorporate findings from the literature that identifies multiple phases during training (Jastrzebski et al., 2020; Shwartz-Ziv & Tishby, 2017a). While often, the performance of language models scales predictably (Kaplan et al., 2020; Srivastava et al., 2022), some tasks instead show breakthrough behavior where a single point along a scaling curve (*i.e.* compute/number of tokens seen/number of parameters versus test loss) shows a sudden improvement in performance (Srivastava et al., 2022; Wei et al., 2022; Caballero et al., 2023). One computational structure, the induction head, emerges in autoregressive language models at a discrete phase change (Olsson et al., 2022) and is associated with handling longer context sizes and in-context learning. In machine translation, Dankers et al. (2022) find a learning progression in which a Transformer first overgeneralizes the literal interpretations of idioms and then memorizes idiomatic behavior. When the training set makes grammatical rules ambiguous, Murty et al. (2023) show that language modelling eventually leads to phase transitions towards the hierarchical version of a rule over an alternative based on linear sequential order. Outside of NLP, phase changes are observed in the acquisition of concepts in strategy games (Lovering et al., 2022; McGrath et al., 2022) and arithmetic (Liu et al., 2022; Nanda et al., 2023). Our work also identifies a specific phase in MLM training, the SAS phase, and analyzes its role in performance and generalization behavior.

We also observe an alignment between phase changes in representational complexity and in generalization performance (Section 4.1), which parallels the observation of Thilak et al. (2022) that generalization time during grokking (Power et al., 2022) aligns with the timing of cycles of classifier weight growth. Lewkowycz et al. (2020) and Hu et al. (2023) report a similar alignment between classifier weight growth and the early transition when loss first begins to decline. Our results indicate that, in natural settings, such phase changes in complexity happen at intermediate points in training when a model first acquires particular representational strategies.

Finally, although we find the same phase transition occurs across multiple training runs, other work indicates that generalization capabilities are sensitive to random seed (Sellam et al., 2022; Juneja et al., 2023; Jordan, 2023). Furthermore, phase transitions may be primarily an artifact of poor hyperparameter settings (Liu et al., 2023), which lead to unstable optimization (Hu et al., 2023). Therefore, it is possible that these abrupt breakthroughs would vanish under the correct architecture and optimizer settings.

## C.2 INTERPRETING TRAINING

Recent work in interpretability has begun to take advantage of the chronology of training in developing a better understanding of models. In some of the first papers explicitly interpreting the training process,

Raghu et al. (2017) and Morcos et al. (2018) use subspace methods to understand model convergence and representational similarity. Adapting their methods, Saphra & Lopez (2019) find that early in training, LSTM language models produce representations similar to other token level tasks, and only begin to model long range context later in training. Liu et al. (2021) use a diverse set of probes to observe that RoBERTa achieves high performance on most linguistic benchmarks early in pre-training, whereas more complex tasks require longer pre-training time.

Some studies find that specific capabilities are often learned in a particular order. In autoregressive language models, Xia et al. (2023) show that training examples tend to be learned in a consistent order independent of model size. In MLMs, Chiang et al. (2020) find that different part of speech tags are learned at different rates, while Warstadt et al. (2020b) find that linguistic inductive biases only emerge late in training. Our work likewise finds that extrinsic grammatical capabilities emerge at a consistent point in training.

While our phase transition results mirror Murty et al. (2022)'s findings that the latent structure of autoregressive language models plateaus in its adherence to formal syntax, their work also finds the structure continues to become more tree-like long after syntacticity plateaus. Their results suggest that continued improvements in performance can still be attributed to interpretable hierarchical latent structure, which may be an inductive bias of some autoregressive model training regimes (Saphra & Lopez, 2020).

Although Appendix I precludes the impact of thresholding effects (Schaeffer et al., 2023; Srivastava et al., 2022) on our results, the relationship between the structure onset and capabilities onset does reflect a dependency pattern similar to the checkmate-in-one task, which Srivastava et al. (2022) consider to be precipitated by smooth scaling in the ability to produce valid chess moves. Even in cases where there is no clear dependency between extrinsic capabilities, there may be internal structures like SAS that emerge smoothly, which can be interpreted as progress measures (Barak et al., 2022; Nanda et al., 2023; Merrill et al., 2023).

### C.2.1 INTERPRETATION THROUGH INTERVENTION

Claims about model interpretations can be subject to causal tests, typically applied at inference time (Vig et al., 2020; Meng et al., 2023). Although causal *training* interventions are rare, some existing work has used them to support claims about interpretable model behavior. Leavitt & Morcos (2020), for example, control the degree of neuron selectivity during training in order to demonstrate that this transparent behavior was often, in fact, maladaptive. Follow-up work (Ranadive et al., 2023) shows that neuron selectivity is only transiently necessary early in training, implying that selective neurons are ultimately vestigial. Likewise, Olsson et al. (2022) modify the Transformer architecture to mimic induction head circuits, in order to test their claim that induction head formation was responsible for an early phase change. In considering the role of SAS in model performance, we likewise intervene during training to support and suppress this behavior.

Our work closely relates to the literature on critical learning periods (Achille et al., 2018), where biased data samples prevent the acquisition of particular features or other model behaviors early in training, leading to a finding that a model which fails to learn certain features early in training cannot easily acquire them later. While those experiments illustrate different phases by removing early features and damaging performance, our experiments elicit *positive* changes by removing certain early behaviors, in order to promote other strategies. Furthermore, the behaviors we suppress early in training are immediately learned as soon as they are permitted, so these phases would not be considered critical learning periods.

In all the preceding experimental work that infers causal relationships, it is possible that some related factor is also affected by the proposed intervention. Our work must likewise confront the possibility of an entangled factor that responds to our intervention. The standard approach to remedy this issue is to intervene in as *targeted* a way as possible, ensuring minimal entanglement between the targeted factor and other factors. Since our approach specifically targets internal syntax structure, we expect that the causal relationships we infer are a direct result of changes in internal syntax representations, even if these representations are not *exactly* SAS but *strongly associated with* SAS. We also observe that the capabilities onset consistently appears after the SAS onset, even as we adjust the timing of the SAS onset to arbitrary points, which supports the connection between SAS—or a closely related

structural pattern—and grammatically capabilities. To the best of our knowledge, this is a novel causal analysis that is enabled by our developmental lens.

## D  BLiMP IMPLEMENTATION DETAILS

BLiMP (Warstadt et al., 2020a) consists of 67 different challenges of 1000 minimal pairs each, covering a variety of syntactic, semantic, and morphological phenomena. To evaluate, we use MLM scoring from Salazar et al. (2020) to compute the pseudoperplexity (PPPL) of the sentences in each minimal pair, defined in terms of the pseudo-log-likelihood (PLL) score:

$$\mathrm{PPPL}(\mathcal{D}) := \exp\left(-\frac{1}{N}\sum_{x\in\mathcal{D}}\mathrm{PLL}(x)\right), \tag{8}$$

where $\mathcal{D}$ is a corpus of text and $N$ is the size of $\mathcal{D}$. Additionally, PLL is defined as

$$\mathrm{PLL}(x) := \sum_{i=1}^{|x|}\log P_{MLM}(x_i|x_{\setminus i};\theta), \tag{9}$$

where $P_{MLM}(x_i|x_{\setminus i};\theta)$ is the probability assigned by the model parameterized by $\theta$ to token $x_i$, given only the context $x$ with the $i$-th token masked out.

The BLiMP accuracy is computed as the proportion of acceptable sentences assigned a higher PLL (or lower PPPL) than the unacceptable alternatives. For example, consider the minimal pair consisting of the sentences "These patients do respect themselves" and "These patients do respect himself," where the former is linguistically acceptable and the latter is not. Suppose BERT_Base assigns the former sentence an average PLL of -0.8, and the latter an average PLL of -6.0. Then we consider BERT_Base to be correct in this case, since the average PLL of the acceptable sentence is higher than the PLL of its unacceptable counterpart.

## E  CORRELATION BETWEEN UAS AND CAPABILITIES

When we consider all 25 MultiBERTs seeds, we can measure the degree to which natural random variation yields a correlation between model quality and implicit parse accuracy (UAS) from SAS. We find tbhat UAS does not correlate with either the MLM test loss ($R^2 = -2821$) or the grammatical capabilities measured by BLiMP ($R^2 = -317$). This complete lack of significant correlation is clear in Fig. 6. Therefore, correlational results do not support the common assumption that SAS leads to grammatical capabilities.

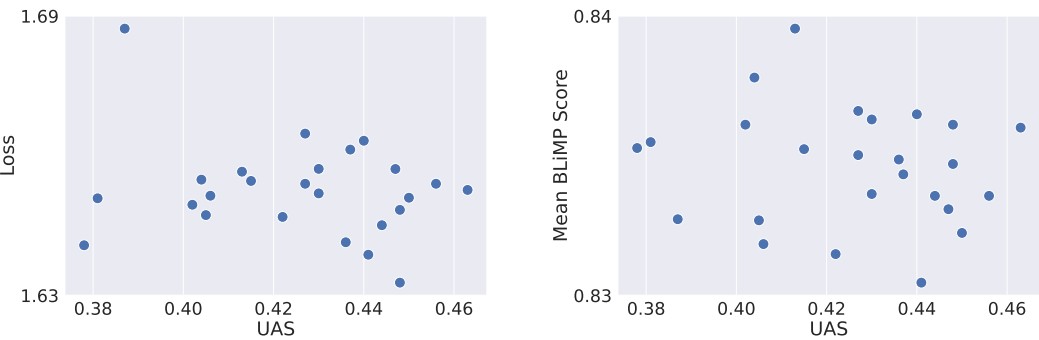

Figure 6: Across 25 MultiBERTs seeds, we do not find a significant correlation between implicit parse accuracy (UAS) and either (a) MLM test loss ($R^2 = -2821$) or (b) grammatical capabilities ($R^2 = -317$).

## F  MULTIBERTS DEVELOPMENTAL ANALYSIS

MultiBERTs (Sellam et al., 2022) is a public release of 25 BERT-base runs, 5 of which have intermediate checkpoints available. Although the intermediate checkpoints are not granular enough to show the timing of the abrupt spike in implicit parse accuracy, or to show a clear break in the accuracy curve for BLiMP, the results nonetheless align with ours (Fig. 7). UAS clearly shows a sharp initial spike followed by a plateau within 20K timesteps. It appears that the BLiMP increase also occurs within the first 20K steps. The loss drops precipitously initially and more slowly after the 20K step checkpoint, as we would expect from the other metrics.

The timing of the UAS plateau implies a slightly faster timeline for the acquisition of linguistic structure compared to our reproduction, and the loss is slightly lower than ours as well, with a slightly higher BLiMP average. Although MultiBERTs appears to be a closer reproduction of BERT with better results compared to our run, we find that it nonetheless is compatible with the same phase transition.

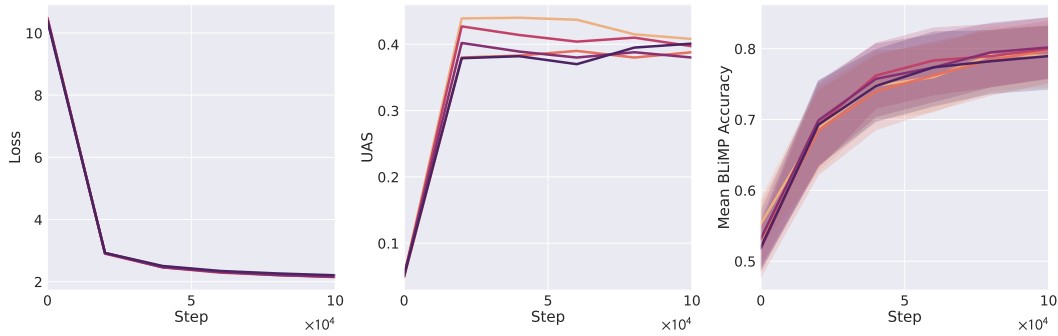

Figure 7: Metrics over the course of training for the 5 MultiBERTs seeds released with intermediate checkpoints. On y-axis: (a) MLM loss (b) Implicit parse accuracy (c) average BLiMP accuracy, with confidence intervals computed across tasks.

## G  COMPLEXITY AND COMPRESSION

Interpretable behaviors such as SAS, by nature of their understandability, must be simplistic. Coincidentally, models tend to learn simpler functions earlier in training (Hermann & Lampinen, 2020; Shah et al., 2020; Nakkiran et al., 2019; Valle-Pérez et al., 2019; Arpit et al., 2017), a tendency often referred to as *simplicity bias*. However, too much simplicity bias can be harmful (Shah et al., 2020) — although simplistic predictors can be parsimonious, we may also lose out on the predictive power of more complex, nuanced features.

If we view SAS as an example of simplicity bias, then we can also view this phase transition through an information theoretic lens. The Information Bottleneck (IB) theory of deep learning (Shwartz-Ziv & Tishby, 2017b) states that the generalization capabilities of deep neural networks (DNNs) can be understood as a form of representation compression. This theory posits that DNNs achieve generalization by selectively discarding noisy and task-irrelevant information from the input, while preserving key features (Shwartz-Ziv, 2022). Subsequent research has provided generalization bounds that support this theory (Shwartz-Ziv et al., 2018; Kawaguchi et al., 2023). Similar principles have been conjectured to explain the capabilities of language models (Chiang, 2023; Cho, 2023; Sutskever, 2023). Current studies distinguish two phases: an initial *memorization* phase followed by a protracted representation *compression* phase (Shwartz-Ziv & Tishby, 2017b; Ben-Shaul et al., 2023). During memorization, SGD explores the multidimensional space of possible solutions. After interpolating, the system undergoes a phase transition into a diffusion phase, marked by chaotic behavior and a reduced rate of convergence as the network learns to compress information.

To validate this theory in MLM training, we analyze various complexity metrics as proxies for the level of compression (see Fig. 2(a) for TwoNN intrinsic dimension (Facco et al., 2017), and

Appendix L.2 for additional complexity/information metrics). Our results largely agree with the IB theory, showing a prevailing trend toward information compression throughout the MLM training process. However, during the acquisition of SAS, a distinct memorization phase emerges. This phase, which begins with the onset of structural complexity, allows the model to expand its capacity for handling new capabilities. A subsequent decline in complexity coincides with the onset of advanced capabilities, thereby confirming the dual-phase nature postulated by the IB theory.

## H  IS THE PHASE TRANSITION CAUSED BY ABRUPT CHANGES IN STEP SIZE?

A possible alternative hypothesis to viewing breakthrough behavior as a conceptual "epiphany" would be that it is an artifact of varying training optimization scales. In other words, there may be some discrete factor in training that causes the optimizer's steps to lengthen, artificially compressing the timescale of learning. The step size decays linearly and the phase transition happens well after warmup ends at 10K steps, meaning that abrupt changes in the hyperparameters are unlikely to be the explanation. However, there may be other factors that affect the magnitude of a step. To confirm that the breakthrough is due to representational structure and not due to a change in the scale of optimization, we consider x-axis scales using a variety of measurements of the progress of optimization. Rather than considering the number of discrete time steps, we consider the following timescales for the weights $w_t$ at timestep $t$:

**Weight magnitude.**  Fig. 8(a) uses the Euclidian distance from the zero-valued origin, which is equivalent to the weight $\ell_2$ norm or $\sqrt{\|w_t\|_2}$.

**Distance from initialization.**  Fig. 8(b) uses the Euclidian distance from the random initialization, i.e., from the weights at timestep 0: $\sqrt{\|w_t - w_0\|_2}$.

**Optimization path length.**  In Fig. 8(c), we approximate the distance traveled during optimization by adding together the lengths of each segment between weight updates, $\sum_{i=1}^{t} \sqrt{\|w_i - w_{i-1}\|_2}$. Because not every timestep is recorded as a checkpoint, we only offer an approximation of the path length by measuring the distance between the recorded sequential checkpoints.

We confirm the phase transitions occur across x-axis scales. Abrupt phase transitions occur whether we consider training timestep, Euclidian distance from initialization, magnitude of the weights, or the path length traveled during optimization.

## I  IS THE PHASE TRANSITION AN ARTIFACT OF THRESHOLDING?

Apparent breakthrough capabilities often become linear when measured with continuous metrics instead of discontinuous ones like accuracy (Srivastava et al., 2022; Schaeffer et al., 2023). Is this the case with our accuracy metrics on SAS and BLiMP?

To the contrary, we find that even using a continuous alternative to the accuracy metric shows similar results. In the case of SAS, providing the attention value placed on the correct target, rather than accuracy based on whether attention is highest for that token, gives a continuous alternative to UAS. In the case of BLiMP, we give the relative probability given to the CLS token on the correct answer in the sequence pair, $p$, compared to the probability $\bar{p}$ of the incorrect CLS token. In other words, the continuous measurement of BLiMP performance is given by $\frac{p}{p+\bar{p}}$. In either case, we see that the phase transition remains clear (Fig. 9).

## J  GLUE TASK ANALYSIS

Fig. 10 shows the GLUE task breakdown while training BERT$_{\text{Base}}$. While not all tasks show a breakthrough in accuracy at the structure onset, most do.

Most GLUE tasks, meanwhile, do not show marked improvements after brief early stage suppression of SAS (Fig. 11). The tasks that have more stability across finetuning seeds show a marked decline in performance as we continue to suppress SH past the alternative strategy onset.

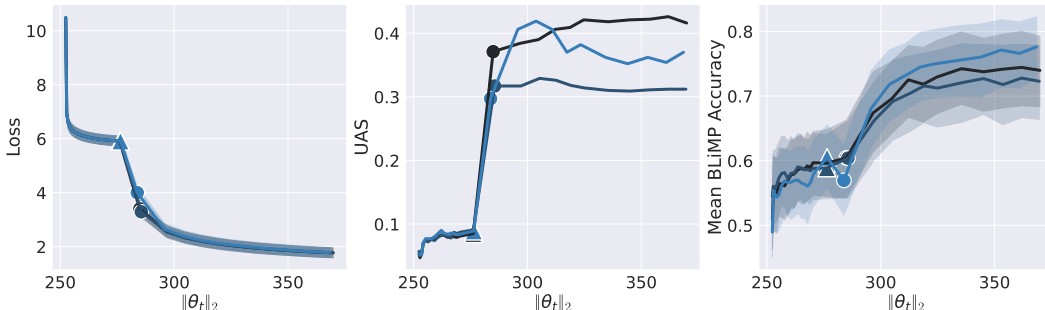

(a) Training scale: Euclidian distance of parameter settings $w_t$ from the zero-valued origin, $\sqrt{\|w_t\|_2}$.

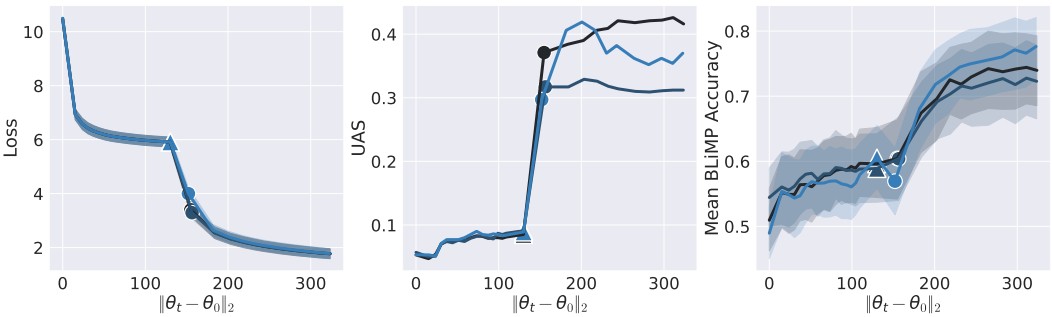

(b) Training scale: Euclidian distance of parameter settings $w_2$ from the model's random initialization, $\sqrt{\|w_t - w_0\|_2}$.

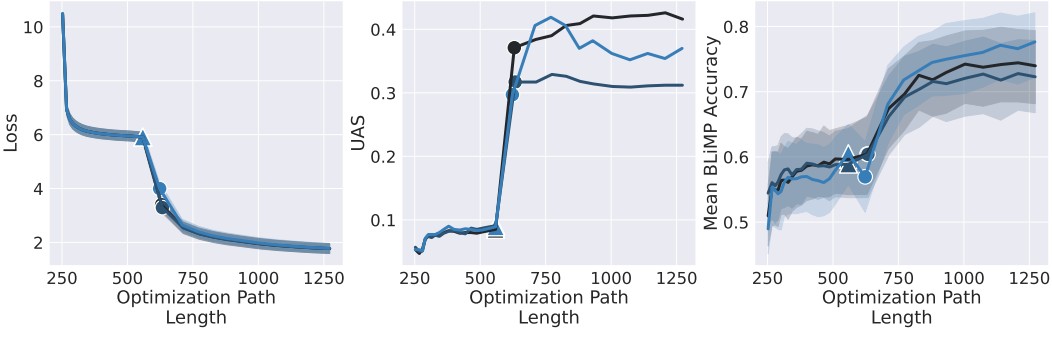

(c) Training scale: Total length of the optimization trajectory after initialization, $\sum_{i=1}^{t} \sqrt{\|w_i - w_{i-1}\|_2}$, with $i$ given only at checkpoint intervals.

Figure 8: Learning trajectories of baseline BERT training, with x-axes reflecting various scales for determining the length of training for checkpoint parameters $w_t$ at time $t$, as an alternative to counting discrete optimization steps. Each curve represents one of three random seeds. Each y-axis corresponds to, from left to right: MLM loss; implicit parse accuracy; and BLiMP average. Structure onset (▲) and capabilities onset (●) are both marked on each line.

## K    BLiMP ANALYSIS

As seen in Fig. 12, most BLiMP tasks show similar responses to multistage SAS regularization: a dip in accuracy for the models that have their regularizer released at the alternative strategy onset and maximum accuracy for a model where the regularizer is released after brief suppression. The model released at the alternative strategy onset has the poorest performance for all tasks except ellipsis.

We note that while training $\mathrm{BERT_{Base}}$, for most BLiMP tasks a clear improvement occurs at the capabilities onset, though intriguingly, for some tasks there is a decline in performance at the structure onset (Fig. 13).

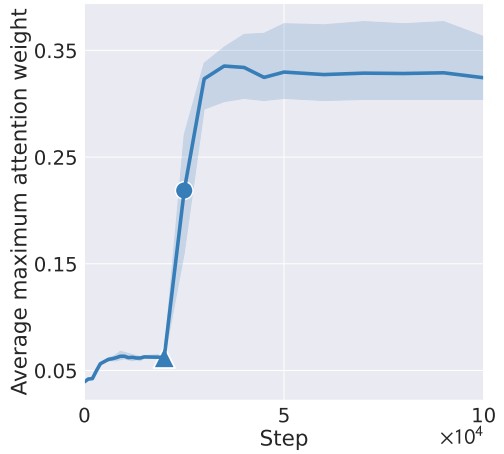
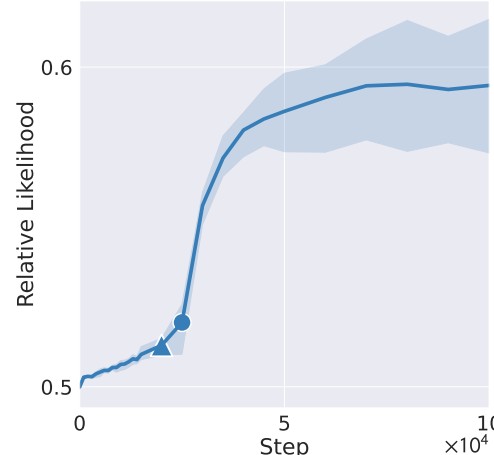

(a) Averaged maximum attention weight on syntactic neighbors, a continuous alternative to UAS accuracy.

(b) Relative likelihood given to the correct member of a minimal pair, a continuous alternative to BLiMP accuracy.

Figure 9: Continuous, non-thresholded metrics for SAS and BLiMP, across three seeds.

## L COMPLEXITY METRICS

The literature on model complexity provides an abundance of metrics which can provide radically different rankings between models (Pimentel et al., 2020). We consider some common complexity metrics during BERT$_{Base}$ training, primarily focusing on intrinsic dimension.

### L.1 INTRINSIC DIMENSION

In order to measure the complexity of the model and its representations, we use **Two-NN intrinsic dimension** (Facco et al., 2017), with other common complexity metrics in Appendix L.2. Two-NN is a fractal measure of intrinsic dimension (ID) that estimates the ID $d$ by computing the rate at which the number of data points within a neighborhood of radius $r$ grows. If we assume that each of the points in the $d$-dimensional ball has locally uniform density, then $d$ can be estimated as a function of the cumulative density of the ratio of distances to the two nearest neighbors of each data point. Two-NN has been used to study the ID of neural network data representations, and can sometimes identify geometric properties that are otherwise obscured by linear dimensionality estimates (Ansuini et al., 2019). In our analyses we compute the Two-NN intrinsic dimension on the [CLS] embeddings of our trained BERT$_{Base}$ models, using pair-wise cosine similarity as our distance metric. We also present the dynamics of several complexity metrics such as the empirical Fisher (EF) and weight norm.

### L.2 OTHER COMPLEXITY METRICS

**Weight magnitude:**   The norm of the classifier weights is an often-studied (Thilak et al., 2022; Lewkowycz et al., 2020) metric for model complexity during training. Shown in Fig. 14(a), this metric rises throughout training, with an inflection up at the capabilities onset. Note that this metric is equivalent to an x-axis scale used in Appendix H.

**Fisher Information:**   Inspired by the approach of Achille et al. (2018), we approximate Fisher Information by $\|\nabla L_{MLM}\|_2^2$, the trend for which is shown in Fig. 14(b). Similar to TwoNN, the model experiences a sharp increase in complexity during SAS acquisition, marking the memorization phase. This phase ends abruptly at the capabilities onset, after which a slow decrease in complexity characterizes the compression phase and improved generalization.

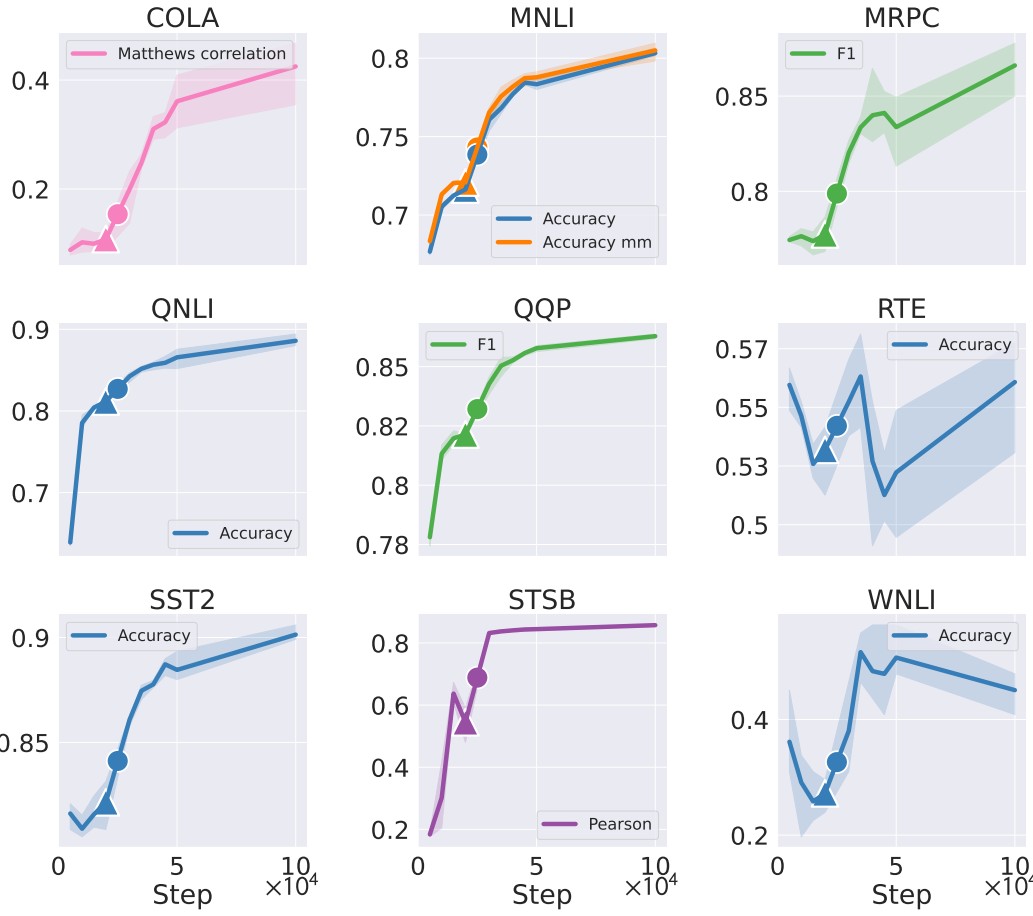

Figure 10: GLUE performance across training for BERT$_{\text{Base}}$, broken down by task. Structure onset (▲) and capabilities onset (●) are marked.

## M WHAT IS THE ALTERNATIVE STRATEGY?

Thus far, we refer to the acquisition of some competing opaque behaviors, defining them only as the useful behaviors *not* supported by SAS. We now argue that an alternative strategy is being learned in the absence of SAS, and characterize it as the use of long range semantic content rather than local syntactic structure.

The first piece of evidence for the use of long-range context is that the onset of the alternative strategy's break in the loss curve coincides with the start of an increase in performance for longer $n$-gram contexts on the task of predicting a masked word within a fixed length context (Fig. 15(b)). For 1000 documents randomly sampled from our validation dataset, we randomly select a segment of $n + 1$ tokens from each document and mask a randomly selected token in each segment. We then compute the average likelihood that the model assigns to the masked token. For example, if $n = 3$ and the randomly selected segment is "a b c d," then we might input the sequence "`[CLS]` a `[MASK]` c d `[SEP]`" to the model and compute the likelihood that the model assigns to the token 'b' at index 2 in the sequence. As training continues, we see spikes in performance for increasingly small contexts while suppressing SAS, over a small window.

When training BERT$_{\text{Base}}$, we also see (Fig. 15(a)) a breakthrough in modeling $n$-gram contexts across lengths during the structure onset. However, we cannot assess whether a similar set of consecutive phase changes occurs in BERT$_{\text{Base}}$, as the difference in timing may occur at a smaller scale than the frequency of saved checkpoints. If all phase transitions in BERT$_{\text{Base}}$ are simultaneous or close to simultaneous, the gradual acquisition of increasingly local structure in BERT$_{\text{SAS-}}$ may account for

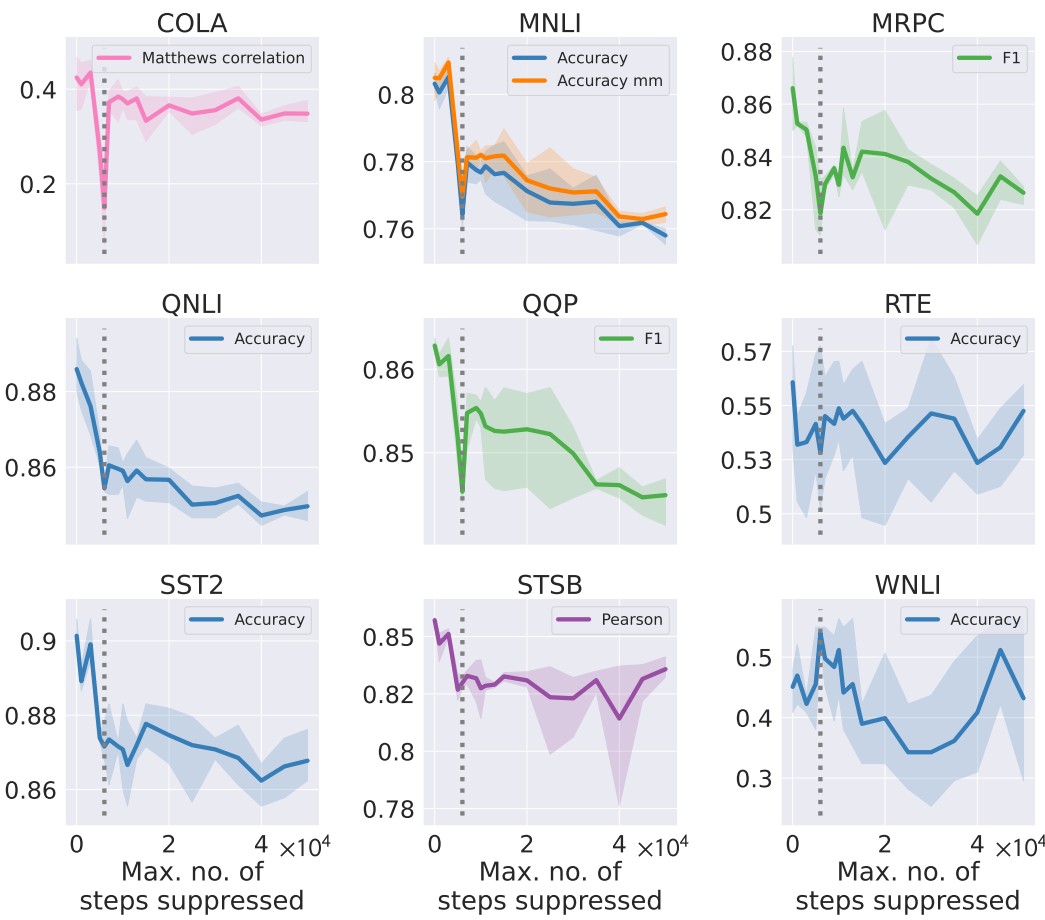

Figure 11: GLUE performance for multi-stage regularized models after 100K timesteps, as a function of the number of steps suppressed and broken down by task. Vertical line marks the BERT$_{SAS-}$ alternative strategy onset.

the more gradual onset of the accompanying loss drop (Fig. 3(a)). The noteworthy difference that we can confirm is that BERT$_{SAS-}$ shows a faster initial increase in performance on n-gram modeling compared to BERT$_{Base}$, suggesting that its break in the loss curve relates to unstructured n-gram modeling, particularly using long range context.

Another piece of evidence is found in the attention distribution. The attention distribution for BERT$_{SAS-}$ is less predictable based on the relative position of the target word (Fig. 16(a)), so unlike in BERT$_{Base}$, the nearest words no longer take the bulk of attention weight. However, on any given sample, the attention distribution is actually higher-entropy for BERT$_{SAS-}$ than BERT$_{Base}$ (Fig. 16(b)), indicating that a small number of tokens still retain the model's focus, although they are not necessarily the nearest tokens. Therefore, BERT$_{SAS-}$ may rely on other semantic factors, and not on position, to determine where to attend. Note that this evidence is weakened by the fact that attention cannot by directly applied as an importance metric (Ethayarajh & Jurafsky, 2021).

## M.1 ONE BIG BREAKTHROUGH OR MANY SMALL BREAKTHROUGHS?

The loss curve after the alternative phase transition under SAS suppression display appears quite different from the baseline after the SAS onset, because the former trajectory declines far more gradually. A clue to this distinction may be in Fig. 15(a), where we see BERT$_{Base}$ exhibit simultaneous breakthroughs in n-gram modeling at every context length. In contrast, BERT$_{SAS-}$ is characterized by a sequence of consecutive breakthroughs starting from the longest context length and gradually reflecting more local context (although this may not account for the difference in transitions—the

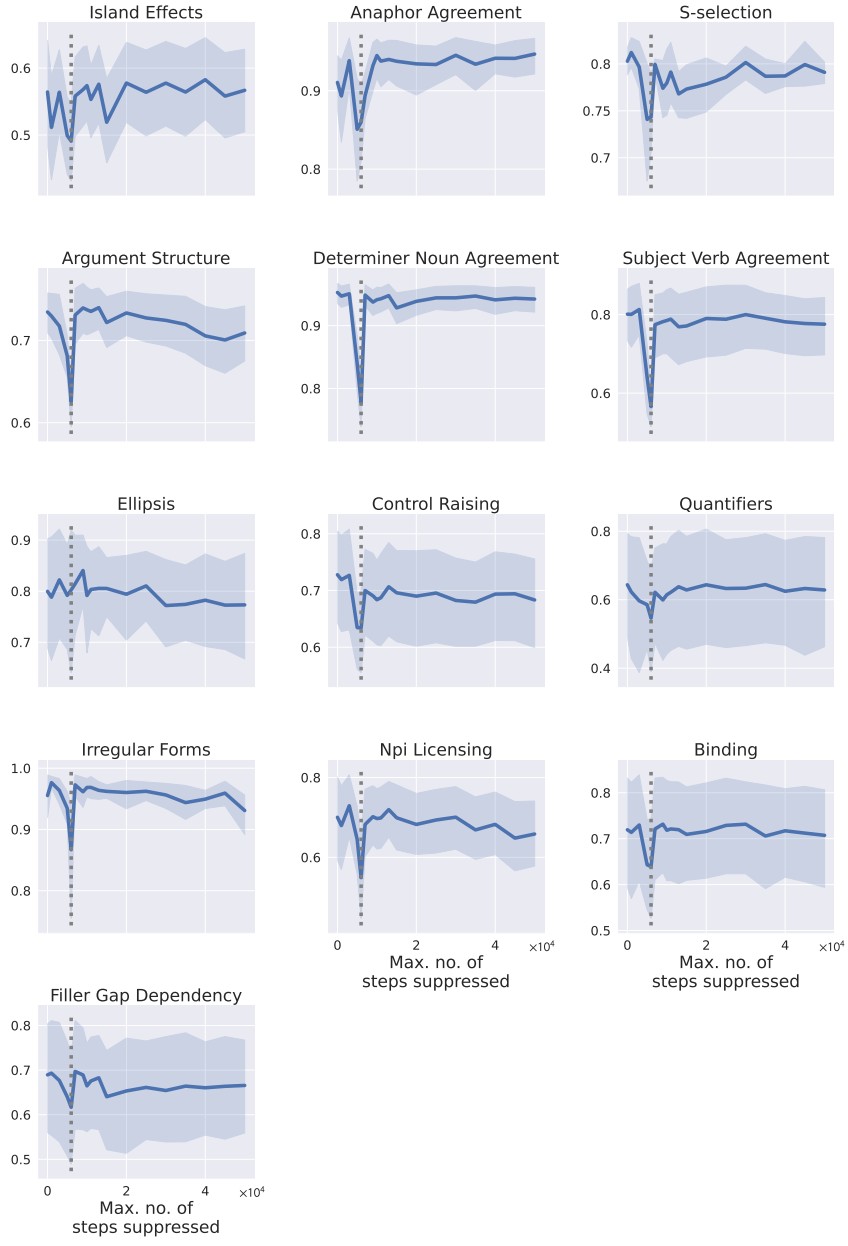

Figure 12: BLiMP accuracy for multistage models at 100k timesteps, broken down by task. Vertical line marks the alternative strategy onset during BERT$_{\text{SAS-}}$ training.

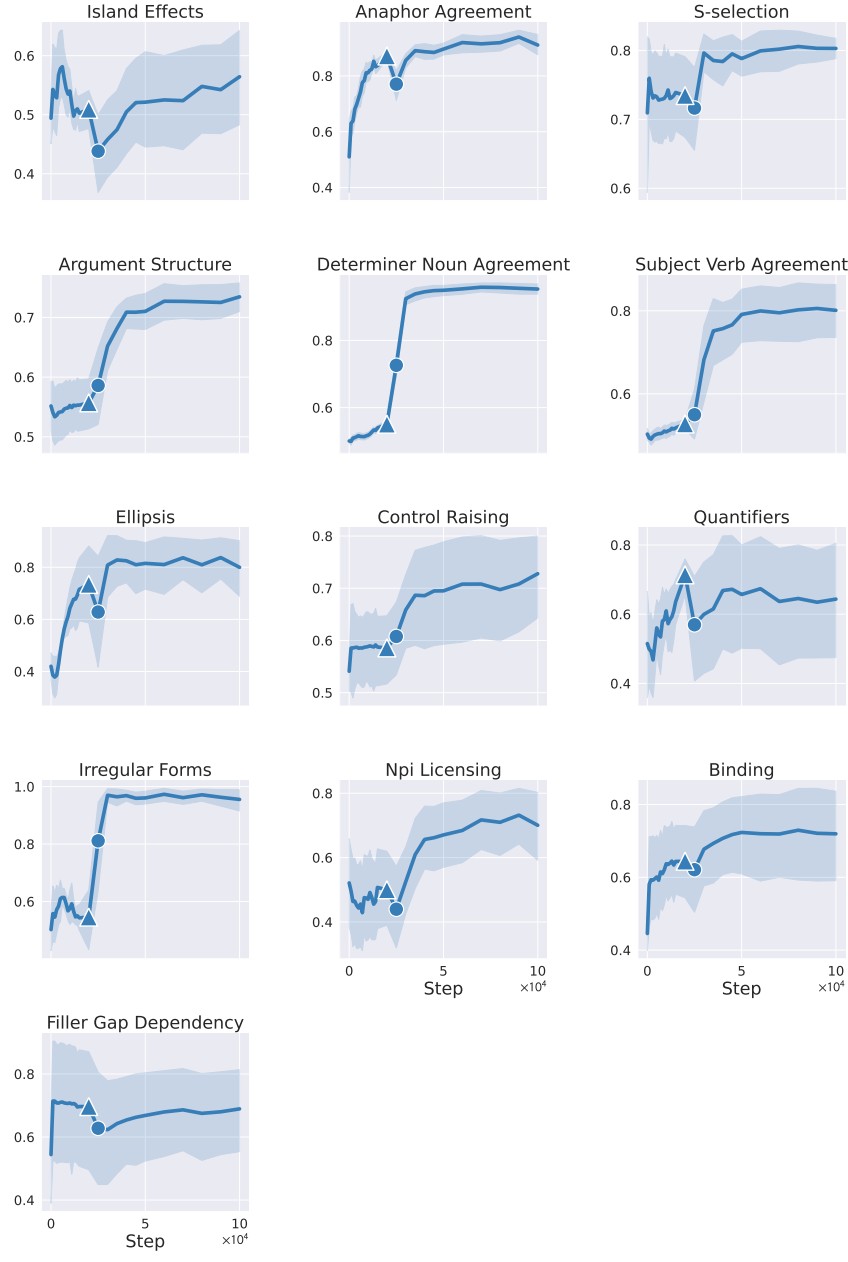

Figure 13: BLiMP accuracy during BERT$_{\text{Base}}$ training broken down by task. Structure and capabilities onsets are marked.

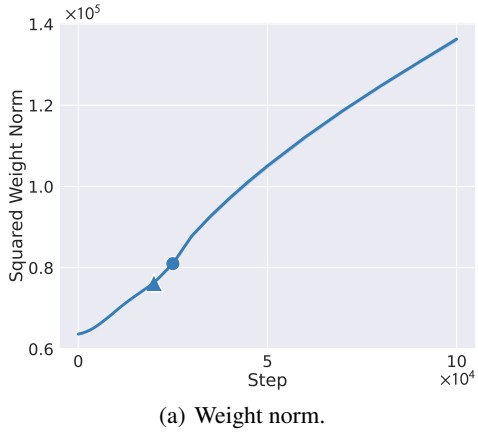

(a) Weight norm.

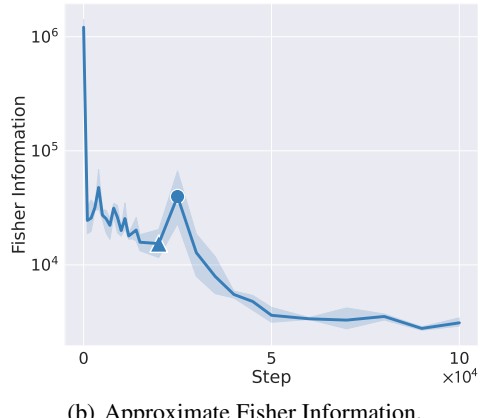

(b) Approximate Fisher Information.

Figure 14: Complexity metrics over time for $BERT_{Base}$. Structure and capabilities onsets are marked.

structure onset in $BERT_{Base}$ happens later in training when the checkpoints are sampled less frequently, and this may account for the apparent simultaneity). Although the performance at the phrase level shows clear phase transitions, the i.i.d. validation loss appears smooth and gradual from the start of the alternative phase transition. This observation is suggestive that a smooth i.i.d. loss curve can elide many phase transitions under various distribution shifts, possibly reflecting the conjecture of Nanda & Lieberum (2022) that "phase transitions are everywhere."

## N    EARLY-SUPPRESSION TRAINING CURVES

Fig. 17 illustrates the general trend that, when we briefly suppress SAS early in training, we can recover and even augment the corresponding UAS spike and loss drop. As we continue to suppress SAS, we lose these benefits and further weaken the transition to SAS. The best timing for hyperparameter release is $BERT_{SAS-}^{(3k)}$, and the training dynamics in Fig. 18 confirm that the multistage approach accelerates and arguments the structure onset and improves model quality during the first 100K steps.

## O    CONTROLLING FOR TIME ALLOWED TO ACQUIRE SAS

Here, we present the same results given in Fig. 4, while controlling for the length of time a model trains after releasing the regularizer in a multistage setting, instead of total training time. We therefore measure performance at exactly 50K steps after setting $\lambda$ to 0, instead of measuring at 100K steps overall. This allows a shorter overall training time which varies across the models. Therefore, the loss of models with shorter first stages improves less compared to the models with longer first stages. Otherwise, the overall patterns remain consistent with the results at a fixed 100K time steps.

## P    LONG TERM IMPACT OF MULTISTAGE SUPPRESSION

To investigate whether the models eventually converge in their biases and structures, we look at functional differences in the form of total variation distance (TVD, or the maximum difference in probabilities that two distributions can assign to the same event) between the output distributions and representational similarity in the form of centered kernel alignment (CKA; Kornblith et al., 2019).

Although the models initially become more similar in their output functions, their distance eventually stabilizes with the average TVD between $BERT_{Base}$ and $BERT_{SAS-}^{(3k)}$ falling above the average between pairs of $BERT_{Base}$ seeds, suggesting that in the absence of another phase transition, the models will remain distinct in their behavior. Meanwhile, the average CKA($BERT_{Base}$, $BERT_{SAS-}^{(3k)}$) similarity diverges during the structure and capabilities onsets but converges again towards the average CKA

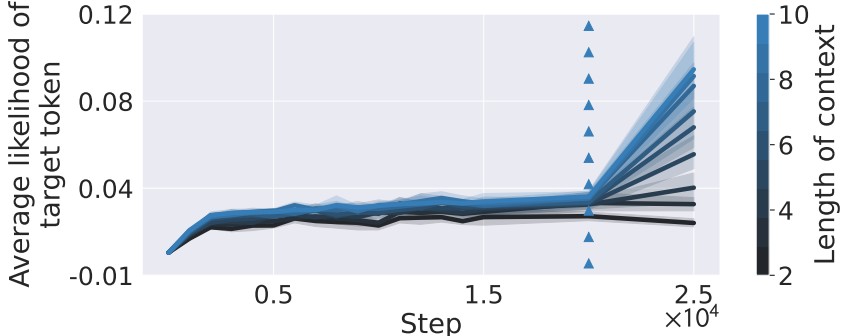

(a) Average BERTBase model likelihood of target token, with varying lengths $n$ of unmasked tokens in its immediate context. Line of triangles (▲) indicates BERTBase's structure onset.

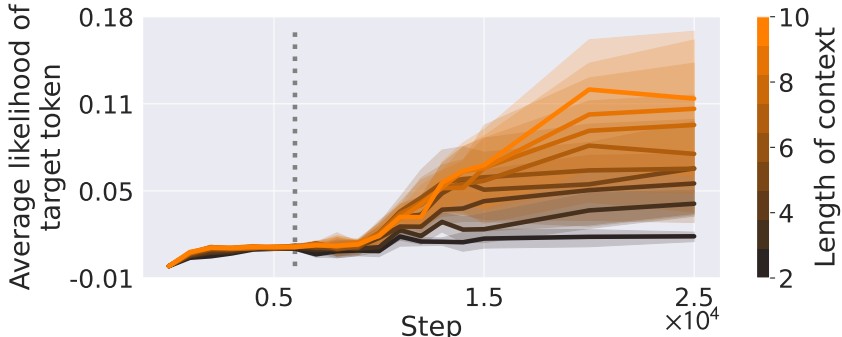

(b) Average BERTSAS- model likelihood of target token, with varying $n$ lengths of unmasked tokens in its immediate context. Dotted line indicates the alternative strategy onset.

Figure 15: The alternative strategy onset, i.e., the break in loss for BERTSAS-, is associated with improvements in n-gram modeling with longer-range contexts. Meanwhile, the structure onset for BERTBase is associated with an improvement in modeling phrases, possibly simultaneously, for all lengths.

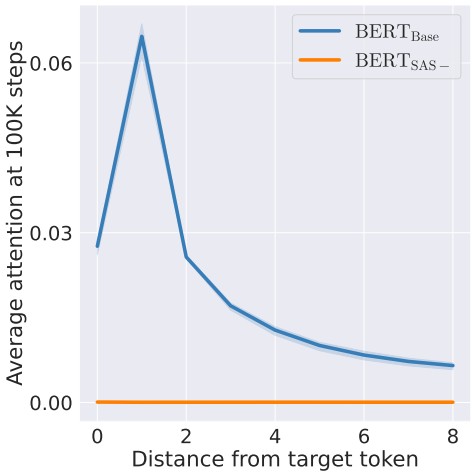

(a) Average attention placed on the target token, as a function of distance (in tokens) from the target token.

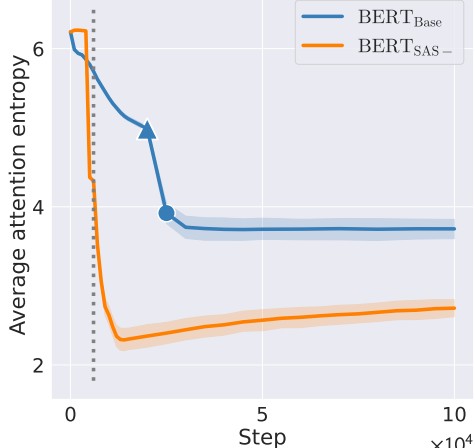

(b) Entropy of the attention distribution, averaged across heads and samples.

Figure 16: The alternative strategy in BERTSAS- is associated with sparser attention compared to the attention distributions in BERTBase. However, the average attention of BERTSAS- (as a function of position) is overall low, indicating that BERTSAS- does not focus attention on nearby tokens as much as BERTBase does, despite the lower entropy.

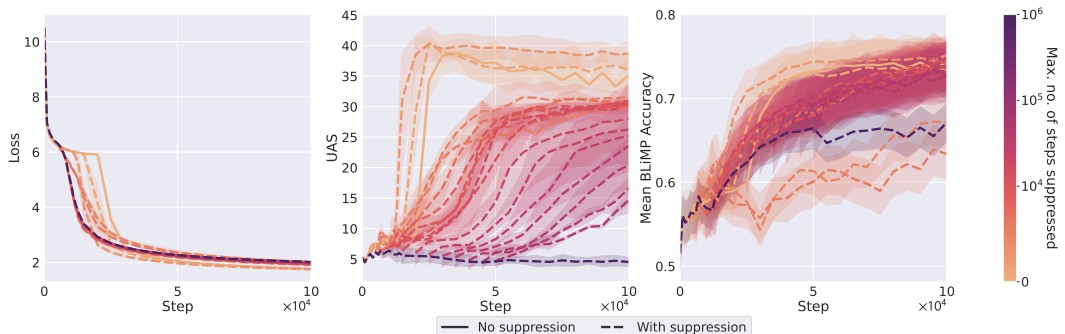

Figure 17: Metrics over the course of training for multistage SAS-regularized models stopped at various points. On y-axis: (a) MLM loss (b) Implicit parse accuracy (c) average BLiMP accuracy. For all multistage training runs, visualized curves begin only after the regularizer is released, i.e., if we suppress SAS for the first 10k steps, the curve begins at 10k. Curve for $BERT_{Base}$ is presented as a solid line, with all suppressed models using dashed lines.

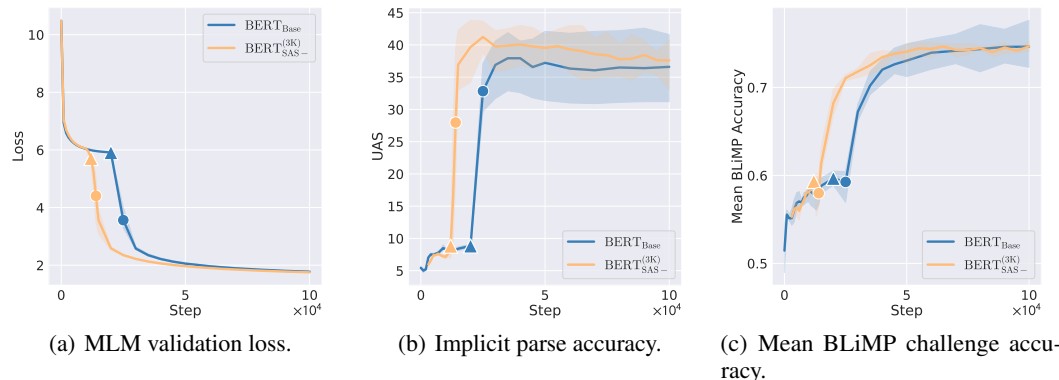

(a) MLM validation loss.  (b) Implicit parse accuracy.  (c) Mean BLiMP challenge accuracy.

Figure 18: Briefly suppressing SAS results in improvements of MLM loss, implicit parse accuracy, and linguistic capabilities early on in training.

between different $BERT_{Base}$ seeds later on. This suggests that even if $BERT_{Base}$ and $BERT_{SAS\text{-}}^{(3k)}$ have high representational similarity, their outputs may still have noticeable differences.

Ultimately, in the long run of training, the difference in quality between $BERT_{SAS\text{-}}^{(3k)}$ and $BERT_{Base}$ ceases to be statistically significant (Tables 1 and 2). While avoiding SAS early in training accelerates convergence and leads to some functional differences in the final models, the critical learning period (Achille et al., 2018) we observe does not continue to hold at scale.

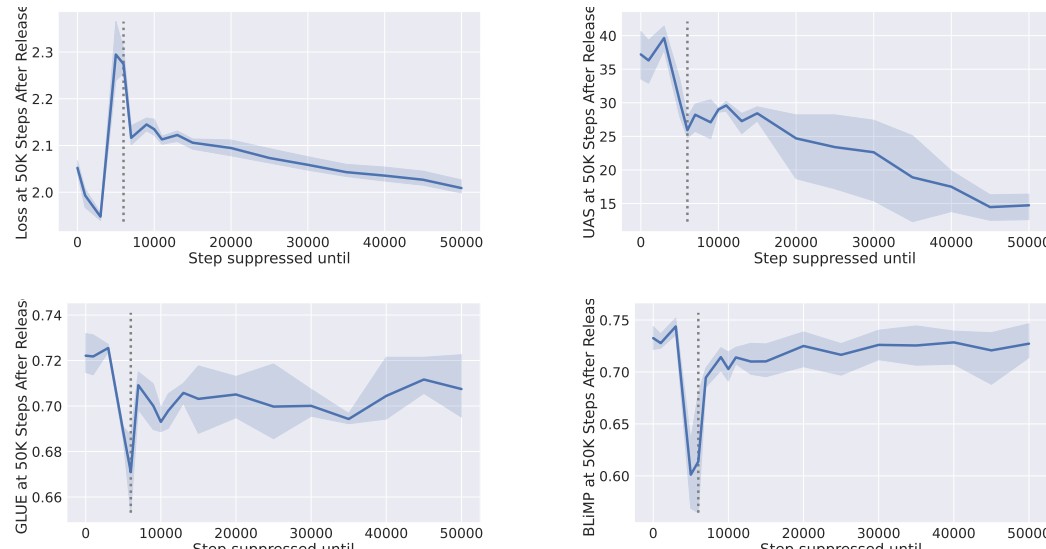

Figure 19: Metrics for the checkpoint 50K steps after the regularizer is removed. X-axis is timestep when regularizer with $\lambda = 0.001$ is removed. On y-axis: (a) MLM loss shown with standard error of the mean across batches (b) Implicit parse accuracy (c) GLUE average (d) BLiMP average. Vertical line marks $\text{BERT}_{\text{SAS-}}$ alternative strategy onset.

Table 2: Evaluation metrics, with standard error, after training for 300K steps ($\sim$ 39M tokens), averaged across three random seeds for each regularizer setting. We selected $\text{BERT}_{\text{SAS-}}^{(3k)}$ as the best multistage hyperparameter setting based on MLM test loss at 100K steps. We selected 300K as the checkpoint to evaluate longer term performance on because longer runs often destabilized, requiring restarts or re-quantization, which force artificial phase transitions late in training.

| | MLM Loss ↓ | GLUE average ↑ | BLiMP average ↑ |
|---|---|---|---|
| $\text{BERT}_{\text{Base}}$ | $\mathbf{1.55 \pm 0.00}$ | $0.74 \pm 0.01$ | $0.75 \pm 0.05$ |
| $\text{BERT}_{\text{SAS+}}$ | $2.17 \pm 0.04$ | $0.63 \pm 0.04$ | $\mathbf{0.77 \pm 0.01}$ |
| $\text{BERT}_{\text{SAS-}}$ | $1.76 \pm 0.02$ | $0.73 \pm 0.00$ | $0.62 \pm 0.05$ |
| $\text{BERT}_{\text{SAS-}}^{(3k)}$ | $\mathbf{1.55 \pm 0.01}$ | $\mathbf{0.75 \pm 0.01}$ | $0.76 \pm 0.02$ |

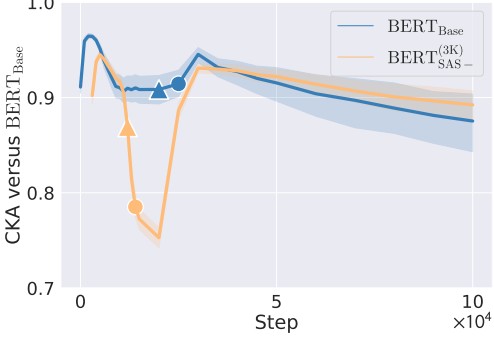

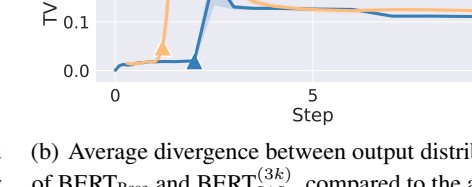

(a) CKA similarity for the activations of $\text{BERT}_{\text{Base}}$ and $\text{BERT}_{\text{SAS-}}^{(3k)}$, compared to the average CKA similarity between different runs of $\text{BERT}_{\text{Base}}$.

(b) Average divergence between output distributions of $\text{BERT}_{\text{Base}}$ and $\text{BERT}_{\text{SAS-}}^{(3k)}$, compared to the average divergence between different runs of $\text{BERT}_{\text{Base}}$.

Figure 20: Representational and functional similarity over time for $\text{BERT}_{\text{Base}}$ and $\text{BERT}_{\text{SAS-}}^{(3k)}$. Structure (▲) and capabilities (●) onsets are marked on each line. Shaded regions are 95% confidence intervals over different pairs of models.

