$\text{BERT}_{\text{Base}}$ and $\text{BERT}^{(3k)}_{\text{SAS-}}$ falling above the average between pairs of $\text{BERT}_{\text{Base}}$ seeds, suggesting that in the absence of another phase transition, the models will remain distinct in their behavior. Meanwhile, the average $\text{CKA}(\text{BERT}_{\text{Base}}, \text{BERT}^{(3k)}_{\text{SAS-}})$ similarity diverges during the structure and capabilities onsets but converges again towards the average CKA between different $\text{BERT}_{\text{Base}}$ seeds later on. This suggests that even if $\text{BERT}_{\text{Base}}$ and $\text{BERT}^{(3k)}_{\text{SAS-}}$ have high representational similarity, their outputs may still have noticeable differences.

Ultimately, in the long run of training, the difference in quality between $\text{BERT}^{(3k)}_{\text{SAS-}}$ and $\text{BERT}_{\text{Base}}$ ceases to be statistically significant (Tables 1 and 2). While avoiding SAS early in training accelerates convergence and leads to some functional differences in the final models, the critical learning period (Achille et al., 2018) we observe does not continue to hold at scale.

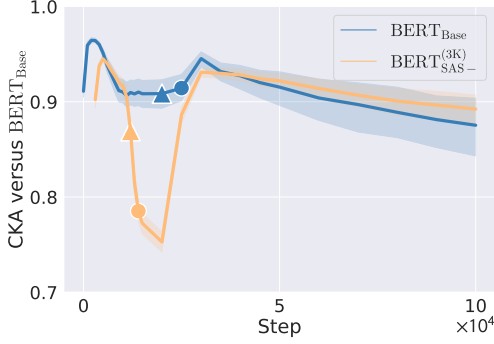 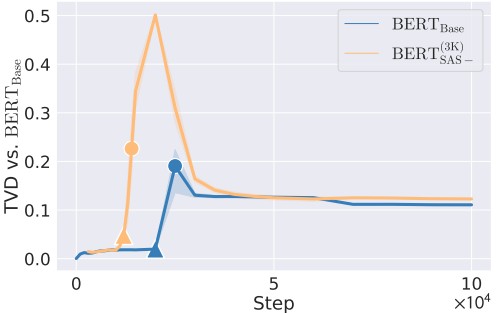

(a) CKA similarity for the activations of $\text{BERT}_{\text{Base}}$ and $\text{BERT}^{(3k)}_{\text{SAS-}}$, compared to the average CKA similarity between different runs of $\text{BERT}_{\text{Base}}$.

(b) Average divergence between output distributions of $\text{BERT}_{\text{Base}}$ and $\text{BERT}^{(3k)}_{\text{SAS-}}$, compared to the average divergence between different runs of $\text{BERT}_{\text{Base}}$.

Figure 20: Representational and functional similarity over time for $\text{BERT}_{\text{Base}}$ and $\text{BERT}^{(3k)}_{\text{SAS-}}$. Structure (▲) and capabilities (●) onsets are marked on each line. Shaded regions are 95% confidence intervals over different pairs of models.