# OpenReview forum: "Sudden Drops in the Loss: Syntax Acquisition, Phase Transitions, and Simplicity Bias in MLMs"
_ICLR.cc/2024/Conference — ICLR 2024 spotlight_

### Official Review · Reviewer_uEHm · 2023-10-21

**Soundness:** 2 fair
**Presentation:** 2 fair
**Contribution:** 2 fair
**Rating:** 5
**Confidence:** 4

**Summary:**

This paper monitors masked language models’ development of syntactic attention structure (SAS) in the attention pattern, grammar capability (measured by the BLiMP dataset) and the GLUE score. They find that

- The model’s grammar capability spikes right after the spike of the model’s SAS score.
- They measure the complexity of the model throughout the pretraining process, and claim that the trend is aligned with the Information Bottleneck Theorem.

They also use a “regularization loss” to interfere with the acquisition of SAS. They find that enhancing/suppressing SAS improves/harms the grammar capabilities of the model.

Though the model is able to develop an “alternative strategy” for acquiring the grammar capabilities when SAS is suppressed, they find that lifting the suppression of SAS during the “phase transition” leads to worse grammar capabilities, while lifting the suppression before the “phase transition” can result in better grammar capabilities. They discuss this phenomenon as related to the simplicity bias of models.

**Strengths:**

1. They empirically show the close relationship between the spike of SAS score, grammar capability, and GLUE score. This is interesting because it may validate the role of understanding syntactic structure for downstream tasks.
2. This work provides many experimental results, which could be useful for better understanding of the model pretraining dynamics.

**Weaknesses:**

# Main concerns:
## 1. The causal relationship between SAS and BLiMP/GLUE scores

I think even though the spike of the BLiMP/GLUE score follows closely after the spike of the SAS score, it is not well substantiated to say that SAS is necessary for the capabilities required for BLiMP and GLUE. The intervention experiment in Sec 4.2, 4.3 can not support the causal relationship between them either. If there is a latent factor X that causes the better SAS, BLiMP, GLUE scores, adding the regularization term may suppress that latent factor X in addition to the SAS score. In this case, suppressing SAS also leads to worse BLiMP and GLUE scores. So that SAS is necessary for BLiMP and GLUE is not the only explanation for the observation in Sec 4.3, 4.3.


## 2. The arguments about the simplicity bias is not clear (to me)

It seems that this paper suggests that SAS indicates that the model suffers from some simplicity bias issue, which is counterintuitive to me. In general I think people use simplicity bias to explain some robustness issues because some spurious (unreliable/non-causal) features are simpler to learn than causal features, or say the model is doing some shortcut learning. However, it is hard to imagine that the syntactic structure is something the model shouldn’t rely on to solve any NLP problem.

I think to talk about simplicity bias, the author should be more clear about the definition of “simplicity” and provide more evidence that the model’s prediction is really “biased” by that specific “simple” feature.


## 3. The motivation of the study

It’s unclear to me why we should look at the development of these “capabilities”. I would like to know how the findings in this paper can potentially direct future research directions?

In general I feel that it’s cool that this paper uses some fancy techniques to show many findings and defines some interesting terminologies. However, it’s unclear to me what the high-level message of this paper is. Unable to capture the coherent theme of this paper, I found it difficult to put all the information in this paper together.

# Minor issues:

1. This paper should be more specific about the definition of “capabilities”.

**Questions:**

## Q1: About section 4.1.1

The authors discuss their findings along with the information bottleneck (IB) theory.

1. It’s unclear to me how the findings agree with what part of the IB theory.
2. It’s also unclear to me how this is related to the findings or the arguments in this paper.

## Q2:  The specific meaning of phase transition in Sec 4.2

In Sec 4.2, the term “phase transition”. Could you clarify what it refers to? Does it refer to the period between the structure onset and the capabilities onset.


## Q3:  The importance of understanding phase transition in general

I understand that *phase transition* is a *hot topic* for some model interpretability community. However, in this work, could you provide more context in which studying *phase transition* is important?

## Suggestions

I understand that every paper needs a reasonable scope to work on and I don’t expect that one single paper explains everything. However, I would suggest that the authors scope this paper more explicitly. “Emergence”, “phase transition” and “capability”, for example, I think are some very general terms, and this paper focuses only on some specific aspects of them. Scoping more clearly and explicitly in the introduction section will help readers (at least me) understand this paper more easily, especially when this paper is discussing MLM models while these terms are usually co-occur with autoregressive language models.

---

> ### Author Response · Authors · 2023-11-21
> **Thank you for your feedback! Revisions are in red and responses are below**
>
> Thank you for volunteering the time and energy to review our paper – we appreciate your detailed feedback on places where our paper could be made more clear. Below are our responses to your concerns and questions.
> - "it is not well substantiated to say that SAS is necessary for the capabilities required for BLiMP and GLUE. The intervention experiment in Sec 4.2, 4.3 can not support the causal relationship between them either. If there is a latent factor X that causes the better SAS, BLiMP, GLUE scores, adding the regularization term may suppress that latent factor X in addition to the SAS score. In this case, suppressing SAS also leads to worse BLiMP and GLUE scores. So that SAS is necessary for BLiMP and GLUE is not the only explanation for the observation in Sec 4.3, 4.3."
>     - Thank you for highlighting these subtleties of causal reasoning. Our regularization explicitly suppresses only the maximum attention weights associated with syntactic structure – if our suppression somehow also suppresses some other latent factor X, then X is likely associated with an internal representation of syntax. Furthermore, our manipulations of SAS consistently impact the timing of both the structure onset and capabilities onset (Figure 5; Figure 18, Appendix N) – *i.e.*, the BLiMP breakthrough always occurs shortly after the UAS spike (if the UAS spike occurs), regardless of whether we accelerate or delay the UAS spike using our SAS regularization. Taken together, our many experiments provide overwhelming evidence that SAS is causally related with the development of complex linguistic capabilities.
>     - It is difficult to completely eliminate the possibility of a latent cause, but our insights about the relationship between the timing of phase transitions and the interactions between different learning strategies hold regardless of whether the underlying cause is exactly SAS or another internal structure that is strongly associated with syntax.
> - "In general I think people use simplicity bias to explain some robustness issues because some spurious (unreliable/non-causal) features are simpler to learn than causal features, or say the model is doing some shortcut learning. However, it is hard to imagine that the syntactic structure is something the model shouldn’t rely on to solve any NLP problem. I think to talk about simplicity bias, the author should be more clear about the definition of “simplicity” and provide more evidence that the model’s prediction is really “biased” by that specific “simple” feature."
>     - Thank you for mentioning this point – we see how it may be unclear. It is true that many studies of simplicity bias employ experiments with heuristics that are not just simplistic, but also spurious. However, generally the definitions of simplicity bias characterize only the complexity of the learner and not its use of spurious features ([Dingle et al.](https://www.nature.com/articles/s41467-018-03101-6), [Pérez et al.](https://arxiv.org/abs/1805.08522), [Nakkiran et al.](https://arxiv.org/abs/1905.11604), [Shah et al.](https://proceedings.neurips.cc/paper/2020/hash/6cfe0e6127fa25df2a0ef2ae1067d915-Abstract.html)). It may so happen that these simplistic heuristics are "shortcuts" and therefore non-causal features, but this is not a requirement for the definition of simplicity bias. Instead, we are merely referring to the observation that *too much* focus on a simplistic strategy (even an effective, non-spurious one) may distract from a more nuanced and complex learning strategy. In some cases, this more complex strategy may simply add more features on top of the simplistic strategy, rather than replacing it.
>     - We have added a brief explanation at the end of Section 4.1 (in red) that also clarifies the definition of simplicity bias – thank you for helping us improve our paper. Appendix C.1 also expands upon the relevant prior work.
>    - As for evidence that the model's prediction is really biased, our aim is only to show that the model's learning is biased towards learning human interpretable artifacts such as SAS, not that the predictions are biased by SAS. We show that BERT-Base naturally learns SAS both without intervention (Fig. 1) and even after limited amounts of suppression (Section 4.3.1).

---

> > ### Author Response · Authors · 2023-11-21
> > **Responses continued**
> >
> > - **Question Q1**:
> >     - Thank you for the feedback – we have updated Section 4.1 (revisions in red) to clarify this point. In short, we are demonstrating how SAS is a phase transition in both the model's structural syntax representation and functional complexity. IB theory simply provides another lens to analyze the phase transitions with – in addition to the emergence of internal syntactic structure followed by the capabilities onset, we also see a brief memorization phase (marked by the increase in functional complexity, Figs. 2(a) and 14(b)) coinciding with the structure onset, followed by an extended compression phase (marked by the gradual decrease of functional complexity, Figs. 2(a) and 14(b)) which starts exactly at the capabilities onset.
> > - **Question Q2**:
> >     - We refer to multiple phase transitions throughout the paper – the structure onset and capabilities onset are phase transitions of the UAS and BLiMP, respectively. We mention this point in the first bullet point in the introduction.
> > - "It’s unclear to me why we should look at the development of these “capabilities”. I would like to know how the findings in this paper can potentially direct future research directions?... **Question Q3**: I understand that phase transition is a hot topic for some model interpretability community. However, in this work, could you provide more context in which studying phase transition is important?"
> >     - Understanding phase transitions guides our understanding of how models acquire the skills necessary for completing complex tasks and highlights potential optimization inefficiencies. These insights can often be used to improve model training – for instance, we show that briefly suppressing SAS can accelerate training (Section 4.3), that training instabilities (or critical points) often occur during the alternative strategy onset (Section 5.2, Figure 4), and that an interpretable dependency exists between internal structure and external capabilities (Section 4.2). In other work, [Achille et al.](https://arxiv.org/abs/1711.08856) identify how data corruption during the early phases of training has a disproportional effect on long-term model performance. [Gur-Ari et al.](https://arxiv.org/abs/1812.04754) identify a phase transition early in training wherein the gradient quickly concentrates within the subspace of the top few eigenvalues of the Hessian; this later inspired techniques for low-dimensional representations of DNN parameters ([Lu et al.](https://dl.acm.org/doi/10.1145/3583780.3614873)) and proofs for the emergence of SGD alignment with outlier eigenspaces ([Arous et al.](https://arxiv.org/abs/2310.03010)) Similarly, detailed analyses of the phase transitions in grokking ([Power et al.](https://arxiv.org/abs/2201.02177), [Thilak et al.](https://arxiv.org/abs/2206.04817)) led to discoveries of how to eliminate grokking (and accelerate generalization) on algorithmic datasets ([Liu et al.](https://openreview.net/forum?id=zDiHoIWa0q1)).
> > - "​​However, I would suggest that the authors scope this paper more explicitly. “Emergence”, “phase transition” and “capability”, for example, I think are some very general terms, and this paper focuses only on some specific aspects of them."
> >     - Thank you for pointing out this potentially confusing language – we understand that these terms have been widely used in the literature to loosely describe many distinct things. We give a more specific definition of what we mean by "capabilities" in the first bullet point of the contributions part of the introduction: "After the spike, the model starts handling complex linguistic phenomena correctly, as signaled by a break in BLiMP score (which we call the **capabilities onset**)." In other words, we refer to "capabilities" as the ability to handle complex linguistic phenomena or challenging grammatical constructions.
> >     - As for the terms "emergence" and "phase transition," we follow the literature ([Wei et al.](https://arxiv.org/abs/2206.07682), [Chowdhery et al.](https://arxiv.org/abs/2204.02311), [Caballero et al.](https://arxiv.org/abs/2210.14891)) and use these terms interchangeably with "break" or "breakthroughs." We precisely define the formula used to identify a breakthrough in Section 2.3 (Equation 2). We have also added a clarification (in red) to that section indicating how we use these terms interchangeably.

---

> ### Comment · Reviewer_uEHm · 2023-11-22
>
> Thanks for your reply.
>
> For the first point, I totally agree your response. I would suggest you make your argument in this way in the paper. Though you claim that it's subtle, **apparently the argument here is different from the main argument in the paper**.
>
> For the second point, after reading your elaboration you added at the end of Section 4.1, I think the argument is much clearer now.
>
> Finally, I think my questions were answered. I appreciate your response.

---

> > ### Author Response · Authors · 2023-11-22
> > **Thank you for your response! Here is our response re: causal inference**
> >
> > Thank you for your response, we appreciate you reading our revisions. We have added a paragraph in Appendix C.2.1 to discuss the possibility of an entangled or similar factor being influenced by our regularizer. However, you did mention that this argument is different from the explanation in the paper. Could you point us to any passage that might contradict our expanded discussion so we can modify it?
> >
> > As we mentioned above, our controlled experimental design still provides strong evidence that internal syntactic structure is causally related with the development of complex linguistic capabilities. Our interventions directly manipulate not only SAS, but also any latent factors. Since our regularization directly targets internal syntactic structure, these latents may be alternative representations of syntax that are not precisely the same as (but strongly associated with) SAS. Thus, our conclusions about internal representations of syntax still hold in this scenario.
> >
> > Generally, the gold standard for experimentally verifying causal relationships is to directly manipulate the independent variable (while keeping all else as controlled as possible) and observing the changes in the dependent variable. As such, our controlled experimental design is standard and also used by many other works that infer causal relationships – [Leavitt and Morcos](https://arxiv.org/abs/2003.01262), for instance, observe changes in test accuracy resulting from regularizing convolutional feature maps to infer a causal relationship between class selectivity and model accuracy. [Liu et al.](https://arxiv.org/abs/2210.01117) directly constrain the weight norm to infer a causal relationship between the "LU mechanism" and grokking.
> >
> > Our intervention is also more *targeted*, in the sense that we target only a specific structural element whereas most interpretability work targets directions or neurons which may be associated with other structural elements. We also show that even if suppressing SAS provides syntax information in some way, this does *not* provide benefits for overall model performance (as evidenced by $BERT_\text{SAS+}$ performing worse than $BERT_\text{Base}$ in Table 1).

---

### Official Review · Reviewer_4MRy · 2023-10-31

**Soundness:** 4 excellent
**Presentation:** 4 excellent
**Contribution:** 3 good
**Rating:** 8
**Confidence:** 4

**Summary:**

The research highlights that understanding model behavior requires observing the training process trajectory, not just analyzing a fully trained model. This study looks into syntax acquisition in masked language models, focusing on the Syntactic Attention Structure (SAS). It shows that SAS emerges suddenly during a specific pretraining phase, marked by a significant loss reduction. Further experiments manipulating SAS confirm its essential role in developing linguistic capabilities, whereas the experiments also find that briefly suppressing SAS improves model quality. The authors explain that SAS competes with other effective traits.

**Strengths:**

- The paper is well-written and easy to follow.
- The research idea and findings (including the appendices) are both intriguing and worthy of being shared with the community.
- The experiments were conducted and executed effectively.

**Weaknesses:**

- I didn't find any major weaknesses, just a few minor questions (detailed below).
- Some individuals might express concerns that the experimental setup is somewhat minimal and may suggest the inclusion of additional elements, such as utilizing RoBERTa or evaluating the model on other, possibly more recent, benchmarks.

**Questions:**

- I would like to know what kind of method/technique is used for encoding positional information. Is it the same as the original positional encoding in Vaswani et al. 2017 (https://arxiv.org/abs/1706.03762) or something more recent variants such as Rotary Positional Encoding (https://arxiv.org/abs/2104.09864v4)? I'm asking this because the syntactic dependency relates to positional information in the sentence. I wonder how much it affects (or does not affect) the experimental setup in this paper.

- This is a more open-ended question but I wonder whether we can observe similar breakthrough (steep drop in loss) in auto-regressive (causal / decoder-only) LMs.

---

> ### Author Response · Authors · 2023-11-21
> **Thank you for your review! Responses are below**
>
> Thank you for your careful review – we appreciate that you also found these ideas interesting and worth sharing with the community. Below are our answers to your questions:
>
> - "I would like to know what kind of method/technique is used for encoding positional information. Is it the same as the original positional encoding in Vaswani et al. 2017 (https://arxiv.org/abs/1706.03762) or something more recent variants such as Rotary Positional Encoding (https://arxiv.org/abs/2104.09864v4)? I'm asking this because the syntactic dependency relates to positional information in the sentence. I wonder how much it affects (or does not affect) the experimental setup in this paper."
>     - We use the same positional embeddings implementation as in the original BERT paper (Devlin et al., https://arxiv.org/abs/1810.04805) – i.e., fully trained absolute sinusoidal positional embeddings. It would definitely be interesting to explore how various positional embeddings affect the emergence of SAS and the subsequent capabilities onset. Although we were unable to re-run our entire suite of experiments with other positional encodings, there does exist limited evidence that language models learn internal syntax representations even without positional embeddings (such as in the ELMo model, in [Hewitt and Manning](https://aclanthology.org/N19-1419/)) and in decoder-only models with causal attention (such as in the GPT-2 model, in [Vig and Belinkov](https://arxiv.org/abs/1906.04284)). These syntax probes are not exactly the same as the one we use, but are likely related (especially the attention-based probe in [Vig and Belinkov](https://arxiv.org/abs/1906.04284)).
> - "This is a more open-ended question but I wonder whether we can observe similar breakthrough (steep drop in loss) in auto-regressive (causal / decoder-only) LMs."
>     - This is an interesting question – there seems to be some empirical evidence that similar breakthroughs exist in auto-regressive LMs. For example, Fig. 2 in [Kaplan et al.](https://arxiv.org/pdf/2001.08361.pdf) shows initial steep loss drops in large decoder-only Transformers, and [Olsson et al.](https://transformer-circuits.pub/2022/in-context-learning-and-induction-heads/index.html#phase-change-more-closely) show that the steep loss drop in shallow attention-only models is associated with a break in in-context learning abilities and induction head formation.

---

> > ### Comment · Reviewer_4MRy · 2023-11-23
> >
> > Thank you for the response and references. They will be helpful for other readers, too.

---

### Official Review · Reviewer_cVcx · 2023-11-01

**Soundness:** 3 good
**Presentation:** 3 good
**Contribution:** 3 good
**Rating:** 8
**Confidence:** 4

**Summary:**

This paper analyzes sudden transitions in the training loss of BERT models, identifying two components to this drop: the development of attention patterns correlated with syntax (SAS), and the subsequent emergence of the ability to make grammaticality judgements. The paper then manipulates SAS via a additional term in the loss and analyzes the effect of these manipulations on the second component. The findings are twofold: (i) acquisition of SAS is a pre-requisite for the grammatical capabilities and (ii) briefly supressing SAS leads to a subsequent increase in grammatical capabilities.

**Strengths:**

This is one of relatively few papers which analyzes learning dynamics in BERT and the transitions identified are intriguing. The paper has the potential of stimulating more work in the same vein, applied to models more advanced than BERT.

**Weaknesses:**

I did not find any major weaknesses in the paper. There is a lot going on, but this is understandable given the novelty of the approach.

That said, some of the framing and terminology could be explicated a bit more carefully. I found the issue of simplicity and simplicity bias especially muddled (see questions below).

**Questions:**

Do I understood correctly you equate the syntax-like attention patterns with simplicity bias, and at some point call them "simple heuristics" (section 5.1)?
This is a bit confusing as in the NLP literature terminology like "simple heuristics" refers to undesirable reliance on surface lexical patterns (like bigrams), and reliance on syntax is considered the opposite of a simple heuristic. It would be good to make sure your unusual framing is not a cause of confusion to the readers.

Minor doubt: since you use WSJ data for testing, why use silver Stanford parser dependencies instead of gold, converted from the manually created trees?

Do you have any inkling of what your mystery alternative strategy may involve?

---

> ### Author Response · Authors · 2023-11-21
> **Thank you for your review! Our revisions are in red and our responses are below**
>
> Thank you for your feedback – we appreciate that you highlighted the novelty of the approach and the potential for stimulating more such work.
>
> - "Do I understood correctly you equate the syntax-like attention patterns with simplicity bias, and at some point call them "simple heuristics" (section 5.1)? This is a bit confusing as in the NLP literature terminology like "simple heuristics" refers to undesirable reliance on surface lexical patterns (like bigrams), and reliance on syntax is considered the opposite of a simple heuristic. It would be good to make sure your unusual framing is not a cause of confusion to the readers."
>     - Thank you for mentioning this potential point of confusion! We have added a brief clarification at the end of Section 4.1 (in red) – it is true that the definition of "simple" is relative, and one might refer to bigram patterns as simple vs. syntax as complex. For our purposes, we refer to any human interpretable artifact as simple, since it must be simple enough to be understood. In addition, syntax patterns can still be considered linguistically simple because it considers only the surface form of language, and not the semantics.
>     - When we refer to "simple heuristics," we are referring only to the relative functional complexity of the strategy, as compared to the complexity of all possible functions that a neural network could encode. This excludes whether the heuristic is a desirable strategy or not. Although many studies of simplicity bias do employ experiments with heuristics that are both simplistic and spurious (and therefore undesirable), generally the definitions of simplicity bias characterize only the complexity of the learner and not its use of spurious features ([Dingle et al.](https://www.nature.com/articles/s41467-018-03101-6), [Pérez et al.](https://arxiv.org/abs/1805.08522), [Nakkiran et al.](https://arxiv.org/abs/1905.11604), [Shah et al.](https://proceedings.neurips.cc/paper/2020/hash/6cfe0e6127fa25df2a0ef2ae1067d915-Abstract.html)). We have also referenced other work on simplicity biases in Appendix C.1.
>  - "Minor doubt: since you use WSJ data for testing, why use silver Stanford parser dependencies instead of gold, converted from the manually created trees?"
>     - Thank you for pointing this out – this is a minor miswording on our part that we have corrected in our paper (in red). For UAS evaluation, we indeed used the manual syntax annotations on WSJ from [PTB-3](https://catalog.ldc.upenn.edu/LDC99T42), but we converted these annotations into Stanford Dependencies format using the [Stanford Dependencies converter](https://nlp.stanford.edu/software/stanford-dependencies.shtml). This was done in order to align with the procedure in [Clark et al.](https://arxiv.org/abs/1906.04341) and the associated [codebase](https://github.com/clarkkev/attention-analysis).
> - "Do you have any inkling of what your mystery alternative strategy may involve?"
>     - We explored a couple hypotheses about the alternative strategy in Appendix M. Unfortunately the evidence did not appear strong enough to include these results in the main text, but we found limited evidence to support the hypotheses that $BERT_\text{SAS-}$ acquires unstructured n-gram statistics sooner than $BERT_\text{Base}$ does (Appendix M, Fig. 15). We also found limited evidence that while $BERT_\text{Base}$ appears to rely heavily on position information early on in training, $BERT_\text{SAS-}$ may rely more on factors other than position (such as semantic information) (Appendix M, Fig. 16).

---

### Official Review · Reviewer_qPGn · 2023-11-02

**Soundness:** 4 excellent
**Presentation:** 3 good
**Contribution:** 4 excellent
**Rating:** 10
**Confidence:** 3

**Summary:**

This paper provides a detailed _developmental_ account of (masked) language models' acquisition of grammatical abilities: the authors measure the the time at which (i) attention heads which pay attention to syntactic structure (dependencies, following prior work showing they emerge) and (ii) grammatical abilities (performance on BLiMP, a linguistic diagnostic set) emerge during training.  In particular, they find that (i) occurs reliably just prior to (ii), though both are abrupt, and that (i) occurs with a _sudden drop in the MLM loss_.  This suggests that the model reliably acquires a certain bit of important latent knowledge (of dependency structure) before behavioral evidence of a skill that uses that knowledge (e.g. grammaticality judgments).  The paper also explores regularization to promote or demote (i), with many, many interesting results about when the sudden drop in loss occurs and how it connects to downstream performance.  This kind of causal intervention shows that MLMs _do_ in fact use syntactic attention heads both when doing masked language modeling and grammaticality judgments, teaching us much more about these phenomena than existing probing methods based on static model artifacts.

**Strengths:**

- Very detailed analysis of the emergence of certain knowledge and skills _across training time_ in a language model.
- Demonstrates that drops in loss correspond to acquisition of syntactic knowledge, which then translates to grammatical performance.
- Methodologically and technically innovative (e.g. regularization as a causal intervention) in a way that moves the state of the probing field forward.
- Extremely wide range of experiments, helping isolate exactly which features of training and measurement matter for this phenomena.  Crucially, they show that these emergence phenomena are not measurement artifacts, since they persist when a discrete scale (training time) is replaced with several continuous ones.

**Weaknesses:**

- All of the results are on a single model architecture (BERT base).  On the one hand, this makes sense, since an extremely wide range of experiments are carried out.  On the other hand, we don't know whether the connection between sudden drops in the loss and syntactic knowledge would apply at larger scales, with causal language modeling, etc.
- There are so many experiments and interesting observations that the main paper makes very frequent reference to a plethora of appendices for more detail.  This makes it a bit hard in places to figure out _exactly what_ is being reported and what it all means.  (E.g. the discussion of the Information Bottleneck was fairly hard to follow, even to someone who knows a bit about that literature.)

**Questions:**

- Fig 1b: why do you think there's so much more variance in the BLiMP results than in the loss curves and UAS scores?

- I'm curious about whether it matters that silver dependencies were used in regularization.  Did you try any other "data-free" regularizers to see if they impact SAS similarly?  E.g. since each token has one head, a regularizer that promotes sparsity of attention should implicitly promote SAS as well and vice versa.

- Missing references: (i) Liu et al 2021, "Probing Across Time": https://aclanthology.org/2021.findings-emnlp.71/ .  (ii) p 4: "Causal methods..." I have an idea of what works the authors have in mind, but think they should be explicitly cited here.

- Will code and data be made publicly available?

---

> ### Author Response · Authors · 2023-11-21
> **Thank you for your review! Revisions are in red and responses are below**
>
> Thank you for your review, we really appreciate that you also found the techniques innovative and the results interesting! Your feedback has also helped us clarify the writing in our paper.
> - "All of the results are on a single model architecture (BERT base). On the one hand, this makes sense, since an extremely wide range of experiments are carried out. On the other hand, we don't know whether the connection between sudden drops in the loss and syntactic knowledge would apply at larger scales, with causal language modeling, etc."
>     - We agree it would be valuable follow-up work to study whether these connections continue to be true with both auto-regressive LMs and larger scales. Although we were unable to re-run our entire suite of experiments using larger or more varied types of language models (due to both time and compute constraints), there does exist limited evidence that causal language models also demonstrate syntactic attention patterns ([Vig and Belinkov](https://arxiv.org/abs/1906.04284)) and that syntax learning improves with scale ([Pérez-Mayos](https://arxiv.org/abs/2109.03160)). Based on these findings, it seems reasonable to speculate that syntactic attention patterns may exhibit a similarly close relationship to the initial loss drop as in BERT-Base. However, confirming this and carefully analyzing the dynamics of this relationship is work we leave for future research.
> - "There are so many experiments and interesting observations that the main paper makes very frequent reference to a plethora of appendices for more detail. This makes it a bit hard in places to figure out exactly what is being reported and what it all means. (E.g. the discussion of the Information Bottleneck was fairly hard to follow, even to someone who knows a bit about that literature.)"
>     - Thank you for this feedback, we have revised the Information Bottleneck section (Section 4.1) to try to simplify and clarify its connection to the rest of the paper. It's true that there are a lot of results in this paper, and we appreciate your feedback on how to clarify it all.
> - "Fig 1b: why do you think there's so much more variance in the BLiMP results than in the loss curves and UAS scores?"
>     - The confidence interval we computed for the BLiMP results in Fig. 1(b) is based on variance across different BLiMP challenges (mentioned in the caption). In general, it is common for models to vary in accuracy across the different challenges ([Warstadt et al.](https://arxiv.org/abs/1912.00582)), since each tests a very specific linguistic capability. The loss curve, on the other hand, has a confidence interval computed from the variance across different examples, which we expect to be lower since the model is solving the same task for each example (masked language modeling). The UAS scores do not have a confidence interval in this particular figure (but in Fig. 3 we include confidence intervals across the three model seeds) since the implementation from [Clark et al.](https://arxiv.org/abs/1906.04341) did not admit straightforward calculations of variance. Since each example may have different syntax relations, it would be difficult to interpret the variance of UAS across different examples.
> - "I'm curious about whether it matters that silver dependencies were used in regularization. Did you try any other "data-free" regularizers to see if they impact SAS similarly? E.g. since each token has one head, a regularizer that promotes sparsity of attention should implicitly promote SAS as well and vice versa."
>     - We did at one point try a *uniform* regularizer that would uniformly regularize any attention weight corresponding to a pair of syntactically related tokens. Although this still required dependency relation labels (and is therefore not "data-free" as you asked about), it would still promote sparsity of attention for $\lambda>0$. However, we found that this uniform regularizer did not suppress SAS as effectively as our proposed regularizer. This is likely because if we try to suppress sparsity with the uniform regularizer, the model can still achieve a low loss with a mostly uniform distribution that spikes only on a single attention weight in a specific head and layer between syntactically related pairs of tokens.
> - "Missing references: (i) Liu et al 2021, "Probing Across Time": https://aclanthology.org/2021.findings-emnlp.71/ . (ii) p 4: "Causal methods..." I have an idea of what works the authors have in mind, but think they should be explicitly cited here."
>     - Thank you, we have added Liu et al. to our related work section in Appendix C.2. We have already cited multiple causal methods in App. C.2.1 but have also added these citations to the first paragraph of Section 4.

---

> > ### Author Response · Authors · 2023-11-21
> > **Responses continued**
> >
> > - "Will code and data be made publicly available?"
> >     - Yes, we will make our code available. Our pretraining uses publicly available datasets ([BookCorpus](https://huggingface.co/datasets/bookcorpus) and [English Wikipedia](https://huggingface.co/datasets/wikipedia)), whereas our evaluation on the WSJ portion of the Penn Treebank requires a membership to LDC, but is available [here](https://catalog.ldc.upenn.edu/LDC99T42).

---

### Author Response · Authors · 2023-11-21
**Thank you for the reviewers' feedback**

We appreciate the thoughtfulness and overall positivity of the reviews, and that all four reviewers acknowledged the strengths of our work. It is gratifying to see the reviewers praise the technical and scientific value: “The research idea and findings (including the appendices) are both intriguing and worthy of being shared with the community” (4MRy); “Methodologically and technically innovative…in a way that moves the state of the probing field forward” (qPGn). We also appreciate that the clarity—“The paper is well-written and easy to follow” (4MRy)—and breadth—“Extremely wide range of experiments” (qPGn)—of our work were highlighted. It is also very reassuring to hear two reviewers state “I did not find any major weaknesses” (cVcx and 4MRy).

The reviews were overall quite positive, though two concerns that emerged across multiple reviewers were that our experiments are limited to the BERT base model, and that the density of experiments and results can make it hard to keep track of everything. With the exception of Reviewer uEHm, who has doubts about the causal power of our method and the motivation for our work, the reviewers’ feedback primarily pertained to minor issues of methodological clarity. We have worked to improve the clarity of the manuscript (with revisions marked in red), and attempt to resolve these concerns in the responses to the individual reviewers below. Overall, we feel that the reviewers’ time and effort have improved the quality of our manuscript and we thank them for this.

---

> ### Comment · Reviewer_cVcx · 2023-11-23
> **Thanks for your response**
>
> After reading the other reviews and the authors' response my view of the paper is largely unchanged. I am keeping my recommendation.

---

### Meta-Review · Area_Chair_PyED · 2023-12-12

**Metareview:**

This is a nice paper that analyzes the training dynamics of MLMs, focusing in particular on the acquisition of various types of syntactic knowledge (measured through the emergence of syntactic attention heads that focus on particular types of structures). The paper has many interesting findings, chief among them that increases in downstream performance of an MLM on the BLIMP dataset are foreshadowed by the emergence of syntactic attention heads, and that such increases are tied to sudden drops in pretraining loss. I think this kind of analysis is very valuable and novel, and it can spur further similar research in autoregressive LLMs. The main criticism of this paper (by Reviewer uEHm) is that the causal connection between the emergence of SAS heads and the BLIMP performance is not as straightforward as the paper makes it seem (e.g., there could be some hidden latent factor that explains the findings just as well). The authors did make some edits to clarify this in response, and regardless of whether or not this latent explanation exists, the reliable timing of these phenomena remains very interesting. Overall, it's a strong paper that deserves acceptance.

**Justification For Why Not Higher Score:**

It's a great paper, but it's a little difficult to see large-scale impact arising from the findings (as uEHm points out, there aren't many concrete directions for improving MLM training based on the results).

**Justification For Why Not Lower Score:**

Most reviewers (and this AC) are strongly positive, and the lone negative reviewer had many of their criticisms addressed during the rebuttal period. I see no reason to reject this paper, and indeed it is worthy of spotlighting at the conference.

---

### Decision · Program_Chairs · 2024-01-16

Accept (spotlight)